# Recovery of neuropsychological function following abstinence from alcohol in adults diagnosed with an alcohol use disorder: Systematic review of longitudinal studies

**Anna Powell** [1,2⊙] *, **Harry Sumnall** [2,3⊙], **Jessica Smith** [2,3‡], **Rebecca Kuiper** [1,2‡], **Catharine Montgomery** [1,2⊙]

**1** School of Psychology, Faculty of Health, Liverpool John Moores University, Liverpool, United Kingdom, **2** Liverpool Centre for Alcohol Research, University of Liverpool, Liverpool, United Kingdom, **3** Public Health Institute, Faculty of Health, Liverpool John Moores University, Liverpool, United Kingdom

⊙ These authors contributed equally to this work.
‡ JS and RK also contributed equally to this work.
* a.powell@2019.ljmu.ac.uk

**Editor:** Valerio Manippa, University of Bari Department of Education Psychology and Communications: Universita degli Studi di Bari Aldo Moro Dipartimento di Scienze della Formazione Psicologia Comunicazione, ITALY

## Abstract

### Background

Alcohol use disorders (AUD) associate with structural and functional brain differences, including impairments in neuropsychological function; however, reviews (mostly cross-sectional) are inconsistent with regards to recovery of such functions following abstinence. Recovery is important, as these impairments associate with treatment outcomes and quality of life.

### Objective(s)

To assess neuropsychological function recovery following abstinence in individuals with a clinical AUD diagnosis. The secondary objective was to assess predictors of neuropsychological recovery in AUD.

### Methods

Following the preregistered protocol (PROSPERO: CRD42022308686), APA PsycInfo, EBSCO MEDLINE, CINAHL, and Web of Science Core Collection were searched between 1999–2022. Study reporting follows the Joanna Briggs Institute (JBI) Manual for Evidence Synthesis, study quality was assessed using the JBI Checklist for Cohort Studies. Eligible studies were those with a longitudinal design that assessed neuropsychological recovery following abstinence from alcohol in adults with a clinical diagnosis of AUD. Studies were excluded if participant group was defined by another or co-morbid condition/injury, or by relapse. Recovery was defined as function reaching 'normal' performance.

**Data Availability Statement:** All relevant data are within the manuscript and its Supporting Information files.

**Funding:** The author(s) received no specific funding for this work.

**Competing interests:** The authors have declared that no competing interests exist.

## Results

Sixteen studies (AUD n = 783, controls n = 390) were selected for narrative synthesis. Most functions demonstrated recovery within 6–12 months, including sub-domains within attention, executive function, perception, and memory, though basic processing speed and working memory updating/tracking recovered earlier. Additionally, verbal fluency was not impaired at baseline (while verbal function was not assessed compared to normal levels), and concept formation and reasoning recovery was inconsistent.

## Conclusions

These results provide evidence that recovery of most functions is possible. While overall robustness of results was good, methodological limitations included lack of control groups, additional methods to self-report to confirm abstinence, description/control for attrition, statistical control of confounds, and of long enough study durations to capture change.

## Introduction

Alcohol use disorders (AUD) are associated with differences and impairments in brain structure and function, including in white and grey matter [1–3], event-related potentials [4], neurotransmitter and metabolic systems [3], and a range of neuropsychological functions, including executive functions, attention, reasoning/abstraction, visuospatial abilities, and verbal and visual short- and long-term memory [5–8]. It is important to understand whether these neuropsychological functional impairments can recover with abstinence, however the literature is inconsistent, with discrepancies between and within review-level work, both regarding the functions that recover, and the duration it takes [5–7]. Specifically, a review of prospective literature found consistent improvement of sustained attention, but inconsistencies for attention, executive function, or memory [7], while two methodologically similar reviews of primarily cross-sectional literature found conflicting results, with one concluding that all assessed domains recovered by a year of abstinence [6], and the other that there was a wide range of continued impairment at this stage [5].

Furthermore, there are methodological issues within this work, with findings derived mostly via cross-sectional research, which limits understanding of pre-existing differences and causality, and with tasks grouped under one function each [5, 6], rather than acknowledging that tasks and their different elements often span various domains and sub-domains [7]. The most suitable prior review is Schulte, Cousijn [7], due to its only including longitudinal studies, and grouping tasks under multiple domains, but even this found inconsistent results, so a more up-to date review is needed. As neuropsychological impairments in AUD can impact treatment outcomes [9], and reduce quality of life [10], research clarifying their recovery is important to inform support throughout treatment and recovery. For the sake of brevity, a more in-depth justification of this review is described elsewhere, in the published protocol [11].

## Objective(s)

The primary objective of this paper was to systematically review the literature to assess neuropsychological function recovery following abstinence in individuals with a clinical AUD diagnosis. The secondary objective was to assess predictors of neuropsychological recovery in AUD.

## Method

The review protocol (which was consistently adhered to) was both registered on PROSPERO (CRD42022308686), and peer-reviewed and published prior to commencement [11]; please see these for a detailed description of review methods, including a template data extraction form. Briefly, four sources (APA PsycINFO, EBSCO MEDLINE, CINAHL, and Web of Science Core Collection), were searched for results from the year 1999 onwards (see S3 Table for search strategies), with hand searching and forward searching also undertaken (see Fig 1 for PRISMA flowchart). Searches were initially run on 10/03/2022 and were re-ran on the 17/03/2023 prior to finalising the synthesis. Longitudinal studies of adults aged 18–64 with a diagnosis of AUD (either mild/moderate/severe via DSM-5, dependence or abuse via DSM-IV, or dependence or harmful use via ICD-10/11), which assessed change in neuropsychological function upon abstinence were included. Change must have been measured using either validated self-report/tasks, or clinical diagnoses/progression of neuropsychological impairment. Baseline could occur before or during active AUD, or in early recovery (a month or less since last used alcohol), and first follow-up could occur from at least two weeks of abstinence onward. A comparator was required, in the form of adults aged 18–64 without AUD, adults with a different severity of AUD, or abstinence duration assessed using regression techniques. Studies were excluded if the population was defined by use of other substances, or another/comorbid condition (including alcohol-related brain injury, major psychiatric condition, head trauma, or relapse). If participants were reported as consuming other substances, alcohol must have been the primary substance (no comorbid substance use disorders), with the exception of individuals with tobacco use disorder (though again, studies were not included if they specifically recruited individuals with AUD who use tobacco).

Screening was undertaken by three assessors, one who screened all data (AP), and two others who each screened half (JS and RK). Inter-reviewer consistency was determined prior to screening and determined to be *good* (AP and RK [κ = .747, *p* < .001]; AP and JS [κ = .641, p < .001]), after which a discussion was held to benchmark criteria. See S4 Table for details of studies excluded at the full-text stage. Upon completion of screening, data extraction was undertaken by AP to obtain authors; year; setting; location; participant demographic characteristics; exact diagnosis; recruitment/follow-up procedures; change in neuropsychological function; data relating to secondary aims (characteristics reported as predictors of neuropsychological recovery). Quality assessment using JBI Checklist for Cohort Studies was independently conducted by RK and KS for 10% of articles, while AP assessed all. A narrative synthesis was conducted due to heterogeneity, which included tabular presentation, a preliminary synthesis, explored relationships in the data, and examined synthesis robustness [12, 13]. The seminal book "Neuropsychological Assessment" by Lezak, Howieson [14] was used to guide synthesis grouping, which included grouping tasks across multiple domains and sub-domains where appropriate (see Table 2 for more information on how tasks were grouped).

## Results

Eighteen studies longitudinally measured neuropsychological recovery from AUD, with follow-ups ranging from 14–18 days of abstinence to 24 months. Three studies were described as using mostly the same cohort [15–17], so the study with the longest follow-up duration at eight months [15] was selected to be the primary source for this cohort. The details of the sixteen studies included in the synthesis can be found in Table 1. Of these, eleven studies compared neuropsychological function to controls (though four of these only tested controls once, and one of these only compared AUD baseline performance to controls, while another only compared AUD follow-up to controls), while two compared to test-provided normative data.

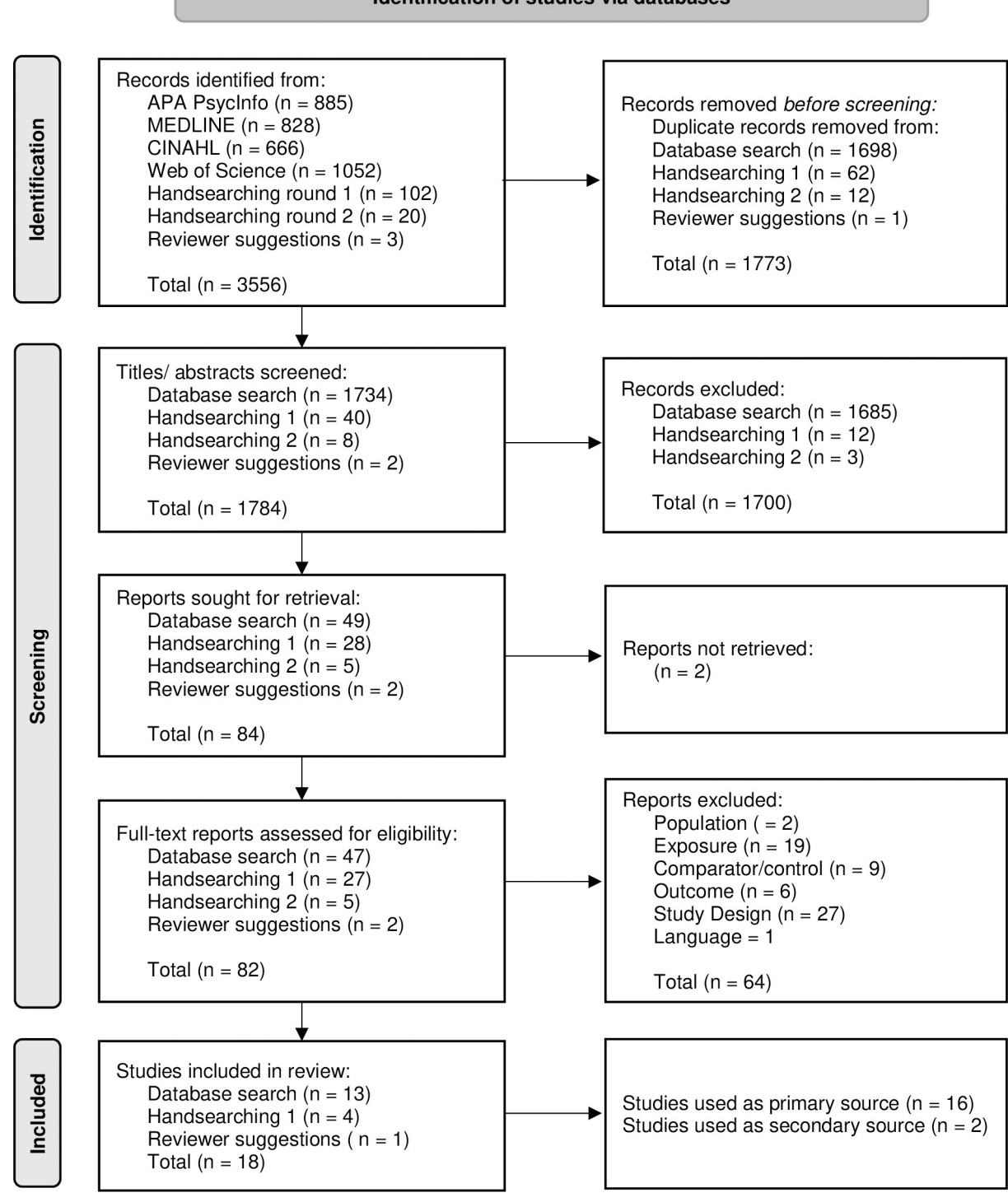

**Fig 1. PRISMA chart of study process.**

The remaining three studies did not compare AUD function to 'normal' performance but did assess the impact of abstinence duration using regression analyses. As a result of these methodologies, it is not possible to exclude the confounding impact of practice effects on

**Table 1. Study characteristics, tasks measured, and outcomes, grouped by study quality (good, moderate, poor).**

| Author, Year | Country, Economic status at time | Setting | Diagnostic tool | % Male | Average age (years) | Timepoints (d = day, w = week, m = month) | Sample size at each time | Comparator | Tasks | Function at T0, impaired vs controls or normative data/not impaired or not compared | Function at follow-up, improvement/no change, or not reached normal performance if compared/ worsened. Significance level p ≤ .05. |
|---|---|---|---|---|---|---|---|---|---|---|---|
| **Good study quality** | | | | | | | | | | | |
| Alhassoon et al., 2012 | USA High | Inpatient | DSM-IV, AD | 100 | 51.4 ± 6.0 | T0 = 2w, T1 = 12m | T0 = 19, T1 = 15 | 15 controls (51.8 ± 7.4 years, 100% Male), assessed twice in same interval. Matched age/education. | TMT-B, DB, HCT | DA, F / **WMU/T** / **F, VCF** | DA, F / **WMU/T** / F, VCF |
| Angerville et al., 2023 | France High | Inpatient | DSM-IV, AD | 75 | 45.5 ± 6.8 | T0 = 8 ± 2d, T1 = 18 ± 2d | T0 = 32, T1 = 32 | 32 controls, and normative data, but no tests of statistical significance to either at T0. Regression methods assessed impact of abstinence duration. | DFR, AS, AVF, HFT | Ver LTM / AC, WMU/T / PS, VF, F / VI | **Ver LTM** / **AC, WMU/T** / **PS, VF, F** / **VI** |
| Bartels et al., 2007 | Germany High | Outpatient | DSM-IV, AD | 71.9 | 44.7 ± 6.2 | T0 = 2-3w, T1 = 3m, T2 = 6m, T3 = 12m, T4 = 24m | T0 = 50, T4 = 32 | Normative data. Three groups; hippocampal dysfunction (HC group) implied by related tasks, no HC dysfunction, additional brain damage, indicated by clinical diagnostics. | CI, Alt, VLT, NVLT, CMT | PS, RI, FA / PS / **Ver LTM** (HC group) / **Vis STM** (HC group) / **Vis LTM** (HC group) | PS, RI, **FA** in HC group by T4 / PS / **Ver LTM** by T3 & T4 (HC group) / **Vis STM** by T3 & T4 (HC group) / **Vis LTM** by T4 (HC group) |
| Czapla et al., 2016 | Germany High | Inpatient | DSM-IV, AD | 81 | 48.05 ± 9.26 | T0 = 2-4w, T1 = 6m | T0 = 94, T1 = 44 (32 of whom relapsed) | 71 healthy controls (46.00 ± 12.02 years, 76% Male), assessed once. Controls only compared at baseline. Regression methods assessed impact of abstinence duration. | RVP, CRT, CGT, AGN, IED | **FA, PS, RI,** WMU/T / PS, RI / **DM** / **RI** / F, **S** | FA, PS, RI, WMU/T / PS, RI / **DM\*\*\*\*** / **RI\*\*\*** / F, S |
| Durazzo et al., 2014 | USA High | 2 + Outpatient | DSM-IV, AD | 84 ns, 85 fs, 96 as | ns = 50 ± 10, fs = 55 ± 13, as = 50 ± 9 | T0 = 1w, T1 = 1m, T2 = 8m | T0 = 93, T1 = 133 (58 added) | 38 never smoking controls from local community (47 ± 9 years, 89% Male), assessed twice in similar interval. Regression methods assessed impact of abstinence duration. Within AUD, never smokers = ns, former smokers = fs, active smokers = as. | SS, DS, DSp, CVL, BVM | **PS** (fs & as) / PS / AC / **Ver STM** (as) / Ver LTM / **Vis STM, Vis LTM** (fs & as) | **PS** by T1 & T2 / **PS** by T1 & T2 (fs & ns, not as) / AC by T1 (ns & as) & T2 / *Ver STM (fs & as), Ver LTM (as) by T1,* Ver STM, Ver LTM by T2 / **Vis STM, Vis LTM** by T2 (fs, not as) |

*(Continued)*

**Table 1.** (Continued)

| Author, Year | Country, Economic status at time | Setting | Diagnostic tool | % Male | Average age (years) | Timepoints (d = day, w = week, m = month) | Sample size at each time | Comparator | Tasks | Function at T0, impaired vs controls or normative data/not impaired or not compared | Function at follow-up, improvement/ no change, or not reached normal performance if compared/ *worsened.* Significance level p ≤ .05. |
|---|---|---|---|---|---|---|---|---|---|---|---|
| Foisy et al., 2007 | Belgium High | Residential post-detox centre | DSM-IV, AD | 57.1 | 42.44 ± 8.05 | T0 = 3-4w, T1 = 3m | T0 = 49, T1 = 22 | 22 controls (44.86 ± 9.31 years, 54.5% Males), tested twice over same interval. Matched age (± 5 years)/ gender/ education. Regression methods assessed abstinence duration impact | EFERp | EFE | EFE |
| Ioime et al., 2018 | Italy High | 2 Outpatient | DSM-IV-TR, AD | 75 | 46.63 ± 8.5 | T0 = 12-17d, T1 = 6m, T2 = 12m | T0 = 41, T1 = 37, T2 = 27 | 40 controls (46.60 ± 6.2 years, 70% Males), tested once. Regression methods assessed impact of abstinence duration. | ST, TMT-B, TMT, B-A, MCST, RPM, RAVL, ROCF, CoF | **FA, RI** **DA, F** **F** **F, S** **VCF** **Ver STM, Ver LTM** **FM, FD, Vis STM, Vis LTM** **FD, FD** | **FA, RI** by T1 & T2 **DA, F** by T1 & T2 **F** by T1 & T2 **F, S** by T1 & T2 VCF Ver STM, *Ver LTM by T1*, Ver LTM by T2 Fm, FD, Vis STM, **Vis LTM** by T2 **FM, FD** by T1 & T2 |
| Loeber et al., 2010 | Germany High | Inpatient | DSM-IV, AD | 60.4 | HD = 47.4 ± 8.4 LD = 44.9 ± 7.7 | T0 = ≥ 5d after end of medication, T1 = 3m, T2 = 6m | T0 = 48, T1 = 35, T2 = 28 | 36 controls (44.4 ± 9.1 years, 56.3% Male), assessed twice in three months. Matched age/ gender/ premorbid intelligence. Grouped by previous detoxes; 2 + detoxes, High-detox = HD, others Low-detox = LD. | IGT, TMT-B + WCST, RAVL + BVRT | M **DA, F** Ver STM, Vis STM | M DA, F by T1, **DA, F** by T2*** Ver STM, Vis STM |
| Petit et al., 2017 | Belgium High | 2 Inpatient | DSM-IV, AD | 73.2 | 49.54 ± 11.58 | T0 = 1d, T1 = 18d | T0 = 41*, T1 = 41* | 41 healthy controls (43.80 ± 11.34 years, 41.5% Male), assessed twice in same interval. Matched age/gender. | ST, DF, B-P | **FA, PS, RI** AC **WMU/T, Vis STM** (10 & 20 sec delays, not 0 or 5 sec) | FA, PS, RI AC **WMU/T, Vis STM** (10 & 20 sec delays) |
| Pitel et al., 2009 | France High | Inpatient | DSM-IV, AD | ** | 47.31 ± 7.42 | T0 = 9.58 ± 4.42d, T1 = 6 m | T0 = 54, T1 = 21 (9 of whom relapsed) | 54 controls (47.27 ± 6.80 years, ** % Male), tested once. Matched age /education. | ST, N-b, Flx, FCSRT | FA **WMU/T, U** F **Ver STM, Ver LTM** | FA **WMU/T** F **Ver STM, Ver LTM** |

(Continued)

Table 1. (Continued)

| Author, Year | Country, Economic status at time | Setting | Diagnostic tool | % Male | Average age (years) | Timepoints (d = day, w = week, m = month) | Sample size at each time | Comparator | Tasks | Function at T0, impaired vs controls or normative data/not impaired or not compared | Function at follow-up, improvement/ no change, or not reached normal performance if compared/ worsened. Significance level p ≤ .05. |
|---|---|---|---|---|---|---|---|---|---|---|---|
| Wegner et al., 2001 | Germany High | Inpatient | ICD-10 & DSM-IV, AD/ misuse | 84.2 | 44.6 ± 2.14 (SEM) | T0 = 0-4d, T1 = 14-18d | T0 = 19, T1 = 19 | 19 controls (42.2 ± 1.75 (SEM), 47.44% Males), tested twice in 3-week span. Matched age/ education. Regression methods assessed impact of abstinence duration. | G/L, DVD | **DA**, Vis STM | **DA**, Vis STM |
| **Moderate study quality** | | | | | | | | | | | |
| Cordovil De Sousa Uva et al., 2010 | Belgium High | Residential | DSM-IV, AD | 48.5 | 48.40 ± 8.2 | T0 = 1-2d, T1 = 14-18d | T0 = 35*, T1 = 35* | 22 controls (44.36 ± 9.64 years, 63.6% Male), assessed twice. Matched age/ gender/ education. IGT performed on split sample, each half did IGT at T0 or T1. | ST, D2, TMT-A, TMT-B, TMT B-A, IGT | **FA, PS, RI** / **FA, PS** / **PS** / **DA, F** / **F** / DM | FA, PS, RI / FA, PS / PS / DA, F / F / DM |
| McCutcheon et al., 2016 | USA High | 3 likely Outpatient | DSM-5, AUD (27 severe, 1 mild) | 0 | 42.3 ± 9.5 | T0 = 27.7 ± 10.2d, T1 = 4 m | T0 = 28, T1 = 18 | No comparison group. Regression methods assessed impact of abstinence duration. | CPT | FA | FA |
| Yeh et al., 2007 | USA High | 2 + Outpatient | DSM-IV, AD/ abuse | 93.3 | ns = 51.5 ± 9.6, s = 47.2 ± 9.8 | T0 = 1w, T1 = 1m, T2 = 7m | T0 = 50, T1 = 46, T2 = 17 | Controls not in statistical model of function recovery. Regression methods assessed impact of abstinence duration. Within AUD, non-smokers = ns, smokers = s. | BVM | Vis STM, Vis LTM | **Vis STM** by T1, **Vis LTM** by T1 (s) & by T2 (ns) |
| **Poor study quality** | | | | | | | | | | | |
| Kaur et al., 2020 | India Lower-middle | Outpatient and Inpatient | ICD-10, AD | 100 | 41.83 ± 9.16 | T0 = 0d, T1 = 1m, T2 = 3m | T0 = 60, T1 = ** | No comparison group. Regression methods assessed impact of abstinence duration. | SSp, L-N, CWA, ANT, WCST, B-G | AC / WMU/T / VF (phonemic) / VF (category) / F, S / FD, FM | **AC** by T1 & T2 / **WMU/T** by T1 & T2 / **VF** by T1 & T2 / **VF** by T1 & T2 / **F, S** by T1 & T2 / **FD, FM** by T2 |

(Continued)

**Table 1.** (Continued)

| Author, Year | Country, Economic status at time | Setting | Diagnostic tool | % Male | Average age (years) | Timepoints (d = day, w = week, m = month) | Sample size at each time | Comparator | Tasks | Function at T0, impaired vs controls or normative data/not impaired or not compared | Function at follow-up, improvement/ no change, or not reached normal performance if compared/ *worsened*. Significance level p ≤ .05. |
|---|---|---|---|---|---|---|---|---|---|---|---|
| Manning et al., 2008 | UK High | Inpatient | ICD-10, AD | 63 | 44.0 ± 7.6 | T0 = 3.8 ± 0.9d<br>T1 = 12m | T0 = 30<br>T1 = 30 | Scores scaled according to age-matched test-provided normative data. Regression methods assessed impact of abstinence duration. | MST<br>SOC<br>L-N<br>CWA<br>ANT<br>IED<br>MR<br>VPA<br>PRM<br>Vc | PS, FM<br>**P**<br>WMU/T<br>VF<br>(phonemic)<br>VF (category)<br>**F, S**<br>VCF<br>Ver STM, **Ver LTM**<br>**Vis LTM**<br>V | PS, FM<br>P<br>**WMU/T**<br>**VF**<br>**VF**<br>F, S<br>**VCF**<br>Ver STM, **Ver LTM**<br>Vis LTM<br>V |

*Stated that relapsers were excluded from study, but these were not described in numbers or characteristics

** data unknown

*** impaired *vs* controls at baseline, but not compared to controls at indicated time

**Table 2. Grouping of neuropsychological functions and tasks.**

| Domain | Sub-domain | Abbreviation | Task (& if pertinent, specific task outcome) |
|---|---|---|---|
| Attention | Focused Attention | FA | CI, ST, D2, CPT, RVP |
| | Divided Attention | DA | TMT-B, G/L |
| | Processing Speed | PS | Alt, ST (congruent RT), TMT-A, MST, D2, CRT, DD, DS, RVP (RT correct, RVP-B), CI (RT), AVF |
| | Attentional Capacity | AC | DF, SSp, DSp, AS |
| Executive Functions | Planning | P | SOC |
| | Decision Making | DM | IGT, CGT |
| | Working Memory Updating/ Tracking | WMU/T | DB, B-P, L-N, N-b, RVP, AS |
| | Response Inhibition | RI | ST (incongruent), commission errors on AGN, RVP, CI & CRT |
| | Verbal Fluency | VF | CWA, ANT, AVF |
| | Flexibility | F | WCST (perseverative errors), MCST (perseverative errors), IED (extra dimensional errors), Flx, TMT-B, TMT B-A, HCT, AVF |
| Concept Formation & Reasoning | Visual Concept Formation | VCF | HCT, MR, RPM |
| | Sort and Shift | S | WCST (perseverative errors, categories achieved), MCST (perseverative errors, categories achieved), IED (extra dimensional errors, stages completed) |
| Learning & Memory | Short-term memory | STM | MF |
| | Verbal short-term memory | Ver STM | VPA (immediate recall), FCSRT (immediate free recall), RAVL (immediate recall), CVL (total recall) |
| | Verbal long-term memory | Ver LTM | VPA (delayed recall), FCSRT (delayed free recall), RAVL (delayed recall), CVL (delayed recall), VLT, DFR |
| | Visual short-term memory | Vis STM | B-P, ROCF (immediate recall), BVM (total recall), BVRT, DVD |
| | Visual long-term memory | Vis LTM | CMT, NVLT, ROCF (delayed recall), BVM (delayed recall), PRM |
| Perception | Figure & Design | FD | B-G, CoF, ROCF (immediate recall) |
| | Emotional Facial Expression | EFE | EFERp |
| | Visual Interference | VI | HFT |
| Verbal Functions | Vocabulary | V | Vc |
| Motor Performance | Fine Motor Function | FM | B-G, MST, CoF, ROCF (immediate recall) |

Tasks: **AGN,** Alcohol Go/No-Go Task; **Alt,** Alertness subtest of Test of Attentional Performance; **ANT,** Animal Names Test; **AS,** Alphabetical Span subtest of the Brief Examination of Alcohol-Related Neuropsychological Impairment; **AVF,** Alternating Verbal Fluency subtest of the Brief Examination of Alcohol-Related Neuropsychological Impairment; B-**G,** Bender-Gestalt Test; **B-P,** Brown-Peterson Technique; **BVM,** Brief Visuospatial Memory Test revised; **BVRT,** Benton Visual Retention Test; **CGT,** Cambridge Gambling Task; **CI,** Crossmodal Integration subtest of Test of Attentional Performance; **CMT,** City Map Test; **CoF,** Copy of Figures; **CWA,** Controlled Word Association (F-A-S); **CPT,** Continuous Performance Test– 2nd Edition; **CRT,** Choice Reaction Time subtest of CANTAB; **CVL,** California Verbal Learning Test revised; **D2,** D2 Cancellation Test; **DB,** Digit Span backward; **DF,** Digit Span forward; **DFR,** Delayed Free Recall subtest of the of the Brief Examination of Alcohol-Related Neuropsychological Impairment; **DS,** Digit Symbol subtest of Wechsler Adult Intelligence Scale– 3rd Edition; **DSp,** Digit Span unspecified (likely composite score of Forwards and Backwards); **DVD,** Delayed Vernier Discrimination; **EFERp,** Emotional Facial Expression Recognition paradigm; **FCSRT,** Free and Cued Selective Reminding Test; **Flx,** Flexibility subtest of Test of Attentional Performance; **FSIQ,** Full-Scale-IQ-2 of Wechsler Abbreviated Scale of Intelligence– 2nd Edition; **G/L,** Global/Local paradigm; **HCT,** Halstead Category Test; **HFT,** Hidden Figure subtest of the Brief Examination of Alcohol-Related Neuropsychological Impairment; **IED,** Intra-Extra Dimensional Set Shift subtest of CANTAB; **IGT,** Iowa Gambling Task; **L-N,** Letter-Number Sequencing; **MCST,** Modified Card Sorting Test; **MR,** Matrix Reasoning subtest of Wechsler Abbreviated Scale of Intelligence– 2nd Edition; **MST,** Motor Screening Test subtest of CANTAB; **N-b,** N-back (2-back); **NVLT,** Nonverbal Learning Test (German version of Kimura Recurring Figures Test); **PRM,** Pattern Recognition Memory subtest of CANTAB; **RAVL,** Rey Auditory Verbal Learning Test; **ROCF,** Rey-Osterrieth Complex Figure Test; **RPM,** Raven's Progressive Matrices; **RVP,** Rapid Visual Information Processing subtest of CANTAB; **SOC,** Stockings of Cambridge subtest of CANTAB; **SS,** Symbol Search subtest of Wechsler Adult Intelligence Scale– 3rd Edition; **SSp,** Spatial Span composite Forwards and Backwards; **ST,** Stroop Colour and Word Test; **TMT B-A,** Trail Making Task part B minus part A; **TMT-B,** Trail Making Task part A; **TMT-B,** Trail Making Task part B; **Vc,** Vocabulary subtest of Wechsler Abbreviated Scale of Intelligence– 2nd Edition; **VLT,** Verbal Learning Test (German version of Recurring Words Test); **VPA,** Verbal Paired Associates subtest of Wechsler Memory Scale– 3rd Edition; **WCST,** Wisconsin Card Sorting Test.

| Domain | Sub-domain | Co10 14-18d | We01 14-18d | Pe17 18d | An23 18d | Ka20 1m | Ye07 1m | Du14 1m | Ka20 3m | Lo10 3m | Ba07 3m | Fo07 3m | Mc16 4m | Pi09 6m | Loe10 6m | Io18 6m | Ba07 6m | Cz15 6m | Ye07 7m | Du4 8m | Ma08 12m | Al12 12m | Io18 12m | Ba07 12m | Ba07 24m |
|---|---|---|---|---|---|---|---|---|---|---|---|---|---|---|---|---|---|---|---|---|---|---|---|---|---|
| **Attention** | Processing Speed | ST, D2, TMT-A | --- | ST | *AVF* | --- | --- | SS, DS | --- | SS, DS | --- | --- | --- | Alt, CI | --- | Alt, CI | --- | CRT, RVP | --- | SS, DS | MST | --- | --- | Alt, CI | Alt, CI |
| | Attentional Capacity | --- | --- | DF | *AS* | SSp | --- | DSp | SSp | --- | --- | --- | --- | --- | --- | --- | --- | --- | --- | DSp | --- | --- | --- | --- | --- |
| | Focused Attention / Divided Attention | ST, D2 TMT-B | G/L | --- | --- | --- | --- | --- | --- | --- TMT-B$ | CI | --- | CPT | ST | --- / *TMT-B*$ | ST / TMT-B | CI | --- | --- | --- | --- TMT-B | --- | ST / TMT-B | CI | *CI* |
| **Executive Functions** | Planning | --- | --- | --- | --- | --- | --- | --- | --- | --- | --- | --- | --- | --- | --- | --- | --- | --- | --- | --- | SOC | --- | --- | --- | --- |
| | Decision Making | IGT | --- | --- | --- | --- | --- | --- | --- | IGT | --- | --- | --- | --- | IGT | --- | --- | *CGT* | --- | --- | --- | --- | --- | --- | --- |
| | Working Memory Updating/Tracking | --- | --- | B-P | *AS* | L-N | --- | --- | L-N | --- | --- | --- | --- | N-b | --- | --- | --- | RVP | --- | --- | *L-N* | DB | --- | --- | --- |
| | Response Inhibition | ST | --- | ST | --- | --- | --- | --- | --- | CI | --- | --- | ST | --- | ST | CI | *AGN* CRT, RVP* | --- | --- | --- | --- | ST | CI | CI | |
| | Verbal Fluency | --- | --- | --- | AVF | *CWA, ANT* | --- | --- | *CWA, ANT* | --- | --- | --- | --- | --- | --- | --- | --- | --- | --- | --- | *CWA, ANT* | --- | --- | --- | |
| | Flexibility | TMT-B, TMT B-A | --- | --- | *AVF* | *WCST* | --- | --- | *WCST* | TMT-B$, WCST$ | --- | --- | --- | Flx | *TMT-B*$, WCST*$ | TMT-B, TMT B-A, MCST | --- | --- | IED | --- | IED | *TMT-B, HCT* | TMT-B, TMT B-A, MCST | --- | |
| **Concept Formation & Reasoning** | Visual Concept Formation | --- | --- | --- | --- | --- | --- | --- | --- | --- | --- | --- | --- | --- | RPM | --- | --- | --- | --- | --- | *MR* | HCT | RPM | --- | --- |
| | Sort & Shift | --- | --- | --- | *WCST* | --- | --- | *WCST* | --- | --- | --- | --- | --- | --- | MCST | --- | IED | --- | --- | --- | IED | --- | MCST | --- | |
| **Learning & Memory** | Verbal short-term memory | --- | --- | --- | --- | --- | --- | CVL | --- | RAVL & | --- | --- | FCSRT | RAVL*& | RAVL | --- | --- | --- | CVL | --- | *VPA* | --- | RAVL | --- | --- |
| | Verbal long-term memory | --- | --- | --- | *DFR* | --- | --- | CVL | --- | VLT | --- | --- | FCSRT | RAVL | VLT | --- | --- | --- | CVL | VPA | --- | RAVL | VLT | VLT | |
| | Visual short-term memory | --- | DVD | B-P | --- | *BVM* | BVM | --- | BVRT & | NVLT | --- | --- | --- | BVRT*& | ROCF | NVLT | --- | *BVM* | BVM | --- | --- | ROCF | NVLT | NVLT | |
| | Visual long-term memory | --- | --- | --- | --- | *BVM* | BVM | --- | CMT | --- | --- | --- | --- | ROCF | CMT | --- | *BVM* | BVM | PRM | --- | --- | ROCF | CMT | CMT | |
| **Perception** | Figure & Design | --- | --- | --- | --- | B-G | --- | --- | *B-G* | --- | --- | --- | --- | --- | ROCF, CoF | --- | --- | --- | --- | --- | --- | ROCF, CoF | --- | --- | --- |
| | Emotional Facial Expression | --- | --- | --- | --- | --- | --- | --- | --- | --- | --- | EFERp | --- | --- | --- | --- | --- | --- | --- | --- | --- | --- | --- | --- | |
| | Visual Interference | --- | --- | --- | *HFT* | --- | --- | --- | --- | --- | --- | --- | --- | --- | --- | --- | --- | --- | --- | --- | --- | --- | --- | --- | |
| **Verbal Functions** | Vocabulary | --- | --- | --- | --- | --- | --- | --- | --- | --- | --- | --- | --- | --- | --- | --- | --- | --- | --- | --- | *Vc* | --- | --- | --- |
| **Motor Performance** | Fine Motor Function | --- | --- | --- | --- | B-G | --- | --- | *B-G* | --- | --- | --- | --- | --- | ROCF, CoF | --- | --- | --- | --- | --- | MST | --- | ROCF, CoF | --- | --- |

\* Impaired vs controls at baseline, but not compared to controls at indicated time; & Composite of verbal/visual STM, individual tests not reported separately; $ Composite of attention/executive function, individual tests not reported separately

--- Not measured  |  Initially impaired vs controls/ normative, no longer impaired  |  No initial impairment vs controls/ normative (or not compared to this), *italics/**bold*** = improved regardless  |  Initially impaired vs controls/ normative, still impaired  |  Performance worsened (and is now impaired if it was not initially)

**Fig 2. Recovery matrix of function over time.**

neuropsychological improvement. The main finding of the review is that sub-domains within attention, executive function, perception, and memory, demonstrate recovery to 'normal' performance levels, generally between 6–12 months. A more in-depth synthesis is now provided.

Impaired function in this review is understood to be that which is significantly worse than control or normative level, while recovery is when function reaches this. Therefore, studies that did not statistically compare to controls or normative performance at baseline and follow-ups have been used to provide an indication of improvement rather than full recovery. Classification of functional domains and sub-domains are displayed in Table 2. A preliminary synthesis was conducted, which involved creating a recovery matrix of all function domains over time (see Fig 2). This informed the narrative synthesis reported below.

## 1.1 Attention

Fourteen studies assessed attention, with follow-ups ranging from 14–18 days of abstinence to 24 months. Of those comparing to 'normal' performance, complete attentional recovery was indicated by twelve months of abstinence at the latest, though there were indications of recovery earlier than this for the sub-domains.

**1.1.1 Processing speed.** Six studies assessed processing speed, with follow-ups ranging from 14–18 days of abstinence to 24 months. Initial impairment was inconsistent, but this may be due to some studies conducting baseline testing too late to capture this. Basic processing speed recovers by one month, but not when other task and goal related elements are involved, such as accuracy on more complex tasks. There is also some indication that verbal processing speed improves faster than visual.

Half of the studies which did not find an initial impairment had generally not conducted baseline testing until at least two weeks of abstinence had already passed [18, 19], except for Petit, Luminet [20], Manning, Wanigaratne [21]. Of those that did find an initial impairment, two conducted baseline assessment within a week of abstinence [15, 22], whilst the third was later at 18 days of abstinence but used a more complex attentional assessment (Rapid Visual Information Processing task, RVP) and recorded reaction time (RT) for correct responses only [19]. Recovery was indicated to occur by a month of abstinence onwards for two basic processing speed tasks [15], Digit Symbol Substitution (DS) and Symbol Search (SS), but processing speed of correct responses on RVP was still impaired by six months [19]. A task with somewhat 'moderate' executive complexity requiring flexibility (Alternating Verbal Fluency, AVF) did display improvement by 18 ± 2 days [23]. Interestingly, this is the only study to have used a verbal task of processing speed, perhaps indicating that this improves faster than visual processing speed. Only Durazzo, Pennington [15] specifically assessed predictors of processing speed recovery. Independently, age and premorbid verbal intelligence predicted change on both tasks in Durazzo, Pennington [15] across the whole sample. Additionally, differential recovery was indicated as a result of smoking status, with active smokers demonstrating the poorest outcomes in both DS and SS (indeed, not recovering to control performance on DS). Furthermore, as a supplementary finding in the same cohort from a secondary source, processing speed recovery in non-smokers was associated with increasing volumes in lobar grey and white matter regions and subcortical regions during 7.5 months of abstinence, though volume increase was similar between smokers and non-smokers, so this is unlikely to explain functional recovery differences between the two groups [16]. Angerville, Ritz [23] did assess predictors of Brief Evaluation of Alcohol-Related Neuropsychological Impairment 'cognitive score', which encompasses processing speed, attentional capacity, working memory updating/tracking, verbal fluency, flexibility, verbal long-term memory, and visual interference. Alcohol Use Disorders Identification Test (AUDIT) scores (which measures alcohol consumption, dependence symptoms, and negative consequences) and age of onset of first alcohol consumption were both significant predictors, with lower age of onset and higher AUDIT score associated with poorer recovery. However due to the composite nature of the predicted outcome, more specific conclusions relating to individual functions cannot be drawn.

**1.1.2 Attentional capacity.** Attentional capacity was assessed by four studies only, and initial impairment and recovery was inconsistent. Follow-ups ranged from 18 days of abstinence to 8 months. It is likely that attentional capacity is not impaired in AUD, and that impairments and improvements in the other studies are more likely working memory performance related.

Petit, Luminet [20] used Digits Forward, a more specific test of capacity [24], finding no impairment or change up to 18 days of abstinence. In contrast, Durazzo, Pennington [15] appeared to use a composite of Digits Forward and Backward, finding impairment, and Kaur, Sidana [25] used a composite of Forwards and Backwards Spatial Span. Composite span measures may be confounded by impairments of working memory [14]), and both indicated improvement, but it is hard to separate this from possible working memory changes. Similarly, Angerville, Ritz [23] used the Alphabetical Span subtest, which involves working memory manipulation, and also found indications of initial impairment (though this was not confirmed using tests of significance), and improvement by 18 days. Furthermore, differences may also be confounded by the modality of span used, as visuospatial and verbal tasks may involve modality-specific processes [26].

Again, age and premorbid intelligence were independent predictors of recovery across AUD [15], and although only never smokers were impaired versus controls at one week of abstinence, only former smokers were at one month. This may indicate a co-occurring impact

of smoking and AUD on attentional capacity (or indeed, working memory) in some patients. As mentioned previously, Angerville, Ritz [23] who used a composite cognitive score which included attentional capacity as one of the contributing functions, found that lower age of onset of first alcohol use and higher AUDIT score associated with poorer recovery.

**1.1.3 Focused attention.** Seven studies assessed focused attention, with follow-ups ranging from 14–18 days of abstinence to 24 months. This was generally initially impaired, which was consistent in early abstinence. Recovery was inconsistent but indicated to occur by 6–12 months in some cases, with discrepancies across baseline and recovery possibly driven by task and methodology differences.

Of the six studies that compared function to normal performance, two found no initial impairment [18, 27], whilst four did [19, 20, 22, 28], one of which [22], used two measures. By 14–18 days of abstinence there was continued impairment [20, 22], while in McCutcheon, Luke [29], who did not recruit controls, performance improved by three months. Performance at six months was inconsistent, with Ioime, Guglielmo [28] finding recovery to control levels on the Stroop, but Czapla, Simon [19] still finding impairment on the RVP. Recovery in Ioime, Guglielmo [28] was maintained at 12 months.

While Pitel, Rivier [27] used the Stroop test, the main outcome was the number of colours named in the interference condition, but other Stroop studies used a combination of incongruent trial RT, and/or isolated incongruent performance from neutral trial performance in some way. These measures may have been more able to comprehensively assess focused attention, as they would be more sensitive to problems with efficiency and executive inhibitory control deficits [30], which are required for focused attention [31]. Indeed, other studies finding impairment in focused attention may have done so due to its reliance on efficiency and inhibitory control, as Continuous Performance Test score in McCutcheon, Luke [29] is a combination of commission errors (response inhibition), and RT correct, while D2 Cancellation Test (used in Cordovil De Sousa Uva, Luminet [22]) assesses speed concurrently with focus, as it is time limited [32]. Furthermore, in Czapla, Simon [19], RVP commission errors (response inhibition) and RT correct (processing speed) were impaired, indicating that there were issues with efficiency and response inhibition, which could possibly have contributed to poor performance on the task overall. This is not supported however by Bartels, Kunert [18], who found no processing speed or inhibitory deficits or change on the task used, but with the latest attention baseline assessment at 2–3 weeks, this finding is less reliable.

When considering the conflicting findings at six months, the RVP task used by Czapla, Simon [19] may arguably involve another layer of functional ability compared to the Stroop (alongside inhibition and efficiency), as it requires a participant to detect target digit sequences by witnessing one digit at a time, so the individual must remember the previous one or two digits (dependent on trial difficulty) to state whether the overall sequence matches that of the target sequence. This involves working memory updating/tracking, which is not required by the Stroop task, perhaps explaining why performance on this task was still impaired.

Regarding predictability, McCutcheon, Luke [29] found that in their women only sample, increases in network drinking scores (drinking behaviour of people important to the individual) between 1–4 months of abstinence, was associated with worsening focused attention, whilst the opposite was true for those whose network drinking decreased.

**1.1.4 Divided attention.** Five studies assessed divided attention, which was generally impaired at baseline, and demonstrated recovery by around six months. Follow-ups ranged from 14–18 days of abstinence to 12 months. Of the five studies, four found initial impairment at baseline [22, 28, 33, 34], while one did not [35]. This impairment improved by six months [28, 33], to the same level as controls [28], which was maintained at 12 months [28]. Improvement to control level was observed very early in Wegner, Günthner [36], by 14–18 days of

abstinence, however compared to the other measures, they used a much simpler divided attention task (a Global/Local paradigm), with current findings suggesting this may not be able to capture continued impairment in AUD.

Alhassoon, Sorg [35], was the exception, finding no impairment vs controls on TMT-B, and no change by 12 months of abstinence, which could be related to the late baseline assessment in this study. Furthermore, the sample in Alhassoon, Sorg [35] were all male, and previous research has demonstrated gender differences on the TMT tasks [37]. Although the exact nature of this relationship is not consistent throughout the literature, at least one study has found that men demonstrate better performance on TMT-B [38].

Predictability was not generally assessed. However, Loeber, Duka [33] found that individuals with two or more previous detoxes performed poorer than those with fewer previous treatments by six months of abstinence on a composite EF/attention score, indicating that repeated cycles of withdrawal and relapse have a damaging influence regarding recovery of divided attention and cognitive flexibility.

## 1.2 Executive functions

Eleven studies assessed aspects of executive function, one of which assessed planning, three decision-making, seven working memory updating/tracking, six response inhibition, three verbal fluency, and nine cognitive flexibility. Despite these abilities being related, the studies demonstrated differential recovery of sub domains. Follow-ups ranged from 14–18 days of abstinence to 24 months.

**1.2.1 Planning.** Planning, assessed using the Stockings of Cambridge task (total score and problem-solving speed), was initially impaired and continued to be so at 12 months of abstinence [21]. Further research is needed to confirm this.

**1.2.2 Decision making.** Decision making was generally impaired at baseline and demonstrated some improvement by six months of abstinence. Follow-ups ranged from 14–18 days of abstinence to 6 months. Interestingly, while Cordovil De Sousa Uva, Luminet [22] and Czapla, Simon [19] found initial impairment, which persisted at 14–18 days of abstinence, Loeber, Duka [33] did not, and found no change by three or six months. Czapla, Simon [19] did not compare to controls beyond baseline but did find some improvement by six months, indicating that when decision making is impaired, improvement can take up to six months.

It is unclear why Loeber, Duka [33] did not find impairment at baseline or change in performance, given that the initial baseline assessments occurred at an abstinence duration comparable to the other two. Loeber, Duka [33] controlled for premorbid intelligence (Vocabulary Test; Schmidt and Metzler [39]) in their analysis, but both Cordovil De Sousa Uva, Luminet [22] and Czapla, Simon [19] reported that controls and AUD participants did not differ regarding educational level, so this is less likely to be the cause of discrepancy, and all three either matched controls on age and gender, or reported no differences in these between the groups. It is possible that another confounding factor that was not assessed or controlled for contributed to the inconsistency.

**1.2.3 Working memory updating/tracking.** Working memory updating/tracking was typically impaired at baseline and demonstrated recovery from as early as 18 days into abstinence, which was generally maintained up to a year of abstinence. Follow-ups ranged from 14–18 days of abstinence to 12 months. The majority of studies found initial impairment (except Manning, Wanigaratne [21]), which typically demonstrated full recovery across studies, including at 18 days [20], six months [27], and 12 months [35], with continued improvement not compared to controls at one and three months [25]. Similarly, Angerville, Ritz [23] also found improvement to control performance level by 18 ± 2 days, though initial deficit was

not confirmed using tests of significance. However, Czapla, Simon [19] found continued impairment at six months.

Despite Manning, Wanigaratne [21] not finding initial impairment compared to normative data, their participants did demonstrate improvement by 12 months, though it is unclear why the initial discrepancy occurred. Regarding predictors, as previously described, Angerville, Ritz [23] found that lower age of onset of first alcohol use and higher AUDIT score predicted poorer recovery of a composite cognitive score which included working memory updating/tracking as one of the contributing functions.

**1.2.4 Response inhibition.** Response inhibition was generally impaired at baseline, and in the majority, demonstrated improvement and in some cases, full recovery between 6–12 months. Follow-ups ranged from 14–18 days of abstinence to 24 months. Five of the studies found initial impairment (except for Crossmodal Integration (CI) commission errors in Bartels, Kunert [18], words recalled in incongruent Stroop in Pitel, Rivier [27], and choice RT in Czapla, Simon [19]). Cordovil De Sousa Uva, Luminet [22] and Petit, Luminet [20] both found that inhibitory control on the Stroop was still impaired around 18 days into abstinence. This recovered to control levels in Ioime, Guglielmo [28] by six months, which was maintained at 12 months. However, while Czapla, Simon [19] found, like Ioime, Guglielmo [28], that response inhibition on an alcohol Go/No-Go (AGN) task improved at six months, commission errors on the CANTAB RVP were still impaired.

As discussed before, as the RVP involves response inhibition and working memory updating/tracking, both of which alone typically demonstrated recovery from six months onwards, this may indicate that when a participant with AUD must perform multiple executive processes at once, performance still suffers compared to controls at this stage. Future research could examine the effect of combined executive processes on performance, as this may be closer to demonstrating real-world executive deficits. Alternatively, this may indicate a difference in the cohort measured by Czapla, Simon [19], but as there was improvement on the AGN this seems less likely (no initial inhibitory deficit on choice RT may be explained by task simplicity [40] relative to the other two examined in this study). Furthermore, the lack of inhibitory deficit on the CI in Bartels, Kunert [18], may relate to research showing that crossmodal stop signals are more effective at prompting response inhibition [41], perhaps due to higher salience [42], so possibly making this task type less able to capture inhibitory impairment in AUD. Finally, issues with using words recalled in the Stroop incongruent condition (as in Pitel, Rivier [27]) have already been discussed in the focused attention section.

**1.2.5 Verbal fluency.** Whilst verbal fluency was not statistically significantly impaired at baseline [21], it did demonstrate improvement consistently across the three studies, at 18 days [23], and one, three, [25] and 12 months [21]. Verbal fluency is often considered an executive function [31], but less consistently than the other measures [43], and it may be more driven by language processing [44], possibly explaining why this did not demonstrate impairment. While not confirmed using tests of significance, Angerville, Ritz [23] did find indications of initial verbal fluency impairment on AVF, however this is a more executive task than the others used as it involves flexibility, which is likely to explain the discrepancy. Furthermore, as mentioned previously, while specific function conclusions cannot be drawn, Angerville, Ritz [23], found that lower age of onset of first alcohol use and higher AUDIT score associated with poorer recovery of a composite cognitive score which included verbal fluency function.

**1.2.6 Flexibility.** Flexibility was generally impaired at baseline, though recovery was inconsistent, with follow-ups ranging from 14–18 days of abstinence to 12 months. The majority found an initial flexibility deficit, except for Czapla, Simon [19] and Alhassoon, Sorg [35], who found impairment on the Halstead Category Test (HCT), but not TMT-B. Impairment was generally consistent across studies during early recovery (between 18 days and three

months) but was inconsistent beyond six months. Loeber, Duka [33], who did not compare to controls beyond baseline, and Ioime, Guglielmo [28] found improvement, even recovery [28] by this stage. However, at 12 months, two studies found continued impairment [21, 35], while individuals in Ioime, Guglielmo [28] maintained their recovery from six months.

Perhaps this suggests that flexibility in some individuals will recover by six months, and that there are predictors of the discrepancies that were not assessed but may also indicate that there is a risk that performance can improve and deteriorate again by 12 months. However, without more studies with multiple follow-ups, this trajectory remains unclear. Interestingly, Alhassoon, Sorg [35], who found maintained impairment at 12 months on the HCT, did not find initial impairment on the TMT-B (which has been discussed previously in relation to gender, in the divided attention section) perhaps indicating that at least in this cohort, the added element of concept formation and reasoning in HCT contributed to initial deficit on this executive measure. Additionally, as an exception, Angerville, Ritz [23] did find improvement by 18 ± 2 days, using AVF. This was the only verbal task assessing flexibility (and processing speed), again perhaps indicating faster improvement of verbal than visual functions.

It is unclear why Czapla, Simon [19] found no impairment on Intra-Extra Dimensional Set Shift (IED) extradimensional shift errors, given that this task is essentially analogous to the WCST and therefore similar to the Modified Card Sorting Test (MCST), and that all other studies using either of these, or even the IED itself, did find impairment [21, 28, 33], however, this was the latest baseline assessment of these, at 2–4 weeks of abstinence, which may have reduced the reliability of the assessment.

As mentioned previously, Loeber, Duka [33], who used a composite TMT-B/WCST measure, found that repeated cycles of abstinence/relapse, worsened outcomes in divided attention and cognitive flexibility by six months. Similarly, while Pitel, Rivier [27] found no initial impairment on the Flexibility task, they did find that individuals who relapsed before the six-month follow-up, then demonstrated worsened performance. Additionally, while specific function conclusions cannot be drawn, Angerville, Ritz [23], found that lower age of onset of first alcohol use and higher AUDIT score predicted poorer recovery of a composite cognitive score which included flexibility.

## 1.3 Concept formation & reasoning

Five studies assessed concept formation and reasoning abilities, three of which assessed visual concept formation, and four the ability to form a concept by which to sort stimuli, and then to switch and form/sort by a new concept. Follow-ups ranged from 1–12 months of abstinence. General concept formation and reasoning skills demonstrated consistent impairment, whilst recovery of sorting and shifting ability was inconsistent.

**1.3.1 Visual concept formation.**   Visual concept formation and reasoning was both initially and consistently impaired, even up to 12 months. Follow-ups ranged from 6 months of abstinence to 12 months. This consistent impairment is logical given that reasoning abilities are often considered a good indicator of premorbid intelligence [14]. The exception was Manning, Wanigaratne [21], who used Matrix Reasoning (MR) combined with Vocabulary to create a Full-Scale IQ score and reported that the IQ score was in the normal range at baseline (but did not describe the range of MR itself), and that MR improved by 12 months. This conflicts with the findings of both Alhassoon, Sorg [35] and Ioime, Guglielmo [28], who found continued impairment on HCT, and Raven's Progressive Matrices, at this abstinence period.

Interestingly, Ioime, Guglielmo [28] did find some improvement by both six and 12 months, but not recovery to control level. Furthermore, grouping of this construct should be considered, as despite it being considered a premorbid ability, some reviews on this topic have

grouped it with functions that would be expected to improve, such as executive function [5, 6]. This may reduce the validity of the synthesis/analysis. Finally, while Schulte, Cousijn [7] concluded that 'Performance IQ' as the function of MR, improved with abstinence in AUD, a closer examination indicates that the only study to find this was Manning, Wanigaratne [21]. Therefore, when additional studies are considered, this seems to be a consistently impaired ability in abstinence from AD.

**1.3.2 Sort and shift.**  The ability to sort and shift was impaired at baseline, and recovery was inconsistent. Follow-ups ranged from 1–12 months of abstinence. Kaur, Sidana [25], who did not compare to controls, indicated that there was some improvement of function on WCST by both one and three months, whilst Ioime, Guglielmo [28] found that MCST performance was fully recovered by both six and 12 months. In contrast, both Czapla, Simon [19] and Manning, Wanigaratne [21] found consistent impairment, at six and 12 months respectively, using IED. It seems that there may be some improvement, but it is unclear if this reaches control performance. Perhaps the modified form of the WCST is less able to monitor continued impairment in AUD, particularly as it simplifies the concept formation [45].

## 1.4 Learning & memory

Memory was assessed by ten studies, four of which assessed verbal STM, six assessed verbal LTM, six assessed visual STM, and five assessed visual LTM. One study only assessed STM as a composite score of visual and verbal ability, which will be discussed separately at the end of this section. Follow-ups ranged from 14–18 days of abstinence to 24 months. Typically, STM for both modalities demonstrated faster recovery than LTM, and despite verbal memory being indicated as recovering faster overall, it was more inclined to worsen during the first six months of abstinence, compared to visual. Visual LTM recovery was the slowest, generally not recovering until two years.

**1.4.1 Verbal short-term memory.**  Verbal STM was generally impaired (except in Manning, Wanigaratne [21]), with recovery occurring from six months onwards in the majority. Follow-ups ranged from 1–12 months of abstinence. Durazzo, Pennington [15] indicated that performance worsened by one month in former and active smokers, but recovered fully by eight, which complements Pitel, Rivier [27] who found full recovery by six months. However, Ioime, Guglielmo [28] found that deficits persisted at both six and 12 months, despite Manning, Wanigaratne [21] finding improvement at 12.

**1.4.2 Verbal long-term memory.**  Verbal LTM was consistently impaired at baseline (except in Durazzo) but recovered in the majority by eight months. Follow-ups ranged from 18 days to 24 months of abstinence. While Angerville, Ritz [23] found improvement by 18 ± 2 days, both Durazzo, Pennington [15] and Ioime, Guglielmo [28] both found worsening of performance, at one and six months respectively, and only in active smokers in the former. There was inconsistency at six months, as alongside Ioime, Guglielmo [28] finding worsening, the cohort in Pitel, Rivier [27] had fully recovered, while Bartels, Kunert [18] found continued impairment, suggesting that impairment is still likely at this stage. Beyond this, full recovery was consistent at eight [15], 12 [18, 21], and 24 months [18], except for Ioime, Guglielmo [28] who still found impairment at 12 months.

It is worth noting that Ioime, Guglielmo [28] was the only study that found impairment in both verbal short- and long-term memory at 12 months, using the Rey Auditory Verbal Learning Test (RAVL) indicating perhaps that this was specific to their cohort. Indeed, it is possible that verbal memory recovery slope/extent is driven by confounding factors. Durazzo, Pennington [15] again found that age and premorbid verbal intelligence independently predicted change across the whole sample, as did education. Additionally, differential rates of

impairment and change were found regarding smoking status, with active/former smokers (and greater lifetime years of smoking) driving initial impairment and showing poorer recovery. Also, both active/former smokers recovered more poorly with increasing age, further highlighting the importance of age as a predictor. Furthermore, as previously mentioned, Angerville, Ritz [23], found that lower age of onset of first alcohol use and higher AUDIT score associated with poorer recovery of a composite cognitive score which included verbal long-term memory.

**1.4.3 Visual short-term memory.** Visual STM was consistently impaired at baseline, recovered by eight months onwards [15, 18, 28] and maintained at 24 months [18]. Exceptions were Wegner, Günthner [36], who used a simple delayed vernier discrimination task and may therefore have been unable to capture group differences, and Petit, Luminet [20], who found recovery on the Brown-Peterson technique by 14–18 days. It is unclear why function was recovered so early in Petit, Luminet [20], however as they also tested controls twice during the same interval, it is unlikely due to practice effects. Follow-ups ranged from 14–18 days of abstinence to 24 months.

**1.4.4 Visual long-term memory.** Visual LTM was consistently impaired at baseline, with recall also consistently impaired at 12 months [18, 28, 35], and signs of complete recovery not evident until 24 months [18]. However Durazzo, Pennington [15], found recovery at eight months (though not in active smokers), and Yeh, Gazdzinski [46], found improvement at one and seven months. Follow-ups ranged from 1–24 months of abstinence.

Once again, age and premorbid verbal intelligence independently predicted change across the whole sample [15]. Both Durazzo, Pennington [15] and Yeh, Gazdzinski [46] found differential recovery as a result of smoking status, with poorest initial performance and outcomes in former and active smokers, the latter of which did not recover to the level of controls [15]. Furthermore, again, both smoking groups demonstrated poorer recovery with increasing age. Yeh, Gazdzinski [46] also investigated brain volume, finding that gains in STM correlated negatively in smokers with brain volume increases during one month of abstinence, which they suggested may indicate that these structural brain changes are pathological.

One extra study that assessed memory did not report memory task outcomes individually. Loeber, Duka [33] created a composite measure of visual and verbal STM (number of pictures remembered on Benton Visual Retention Test, and words remembered on RAVL). This study was an outlier, as it did not find initial impairment, or change of this memory function across three or six months, perhaps indicating that a composite measure is less valid.

## 1.5 Perception and motor performance

Only five studies assessed visual perception, three of which also assessed figure and design reproduction (drawing, an indicator of fine motor function). One study measured visual interference, and one further study assessed fine motor function alone, which is synthesised here also. Follow-ups ranged from 18 days to 12 months of abstinence. It seems that fine motor function and perception of simple designs and visual interference generally responds well to abstinence, improving or even recovering by 6 months. In contrast, perception of more complex designs requires up to 12 months to recover.

Indeed, the earliest improvement was perception of visual interference, which had improved by 18 ± 2 days [23], while recognition and reproduction of complex figures was still impaired by six months but did recover by 12 [28]. All other tasks involved copying much simpler designs (perception and fine motor performance), which showed recovery by six and 12 months [28], or demonstrated improvement consistently at one and three months [25]. Furthermore, Motor Screening Test performance was not impaired at a baseline assessment of

two weeks [21]. Emotional expression recognition was impaired in both accuracy of emotion judgement, and judgement of emotion intensity, in AUD versus controls [47], which did not recover by three months of abstinence. Due to the lack of studies assessing emotional decoding, it is difficult to synthesise this finding, and it is unclear how long recovery would take. Regarding predictors, while specific function conclusions cannot be drawn, Angerville, Ritz [23], found that lower age of onset and higher AUDIT score associated with poorer recovery of a composite cognitive score which included visual interference.

### 1.6 Verbal function

Only one study assessed verbal knowledge [21], finding an improvement by one month of abstinence. While this was not directly compared to normal performance, it was combined with MR to form an IQ measure, which was in the normal range at baseline. It is not possible to synthesise much from this given the single study, however these results show that at least in this cohort, vocabulary was likely unimpaired in early abstinence, but demonstrated improvement regardless.

## Discussion

The aim of this review was to examine recovery of neuropsychological function following abstinence in AUD, with the expectation being that every domain assessed would likely be impaired upon baseline testing, but that recovery would differ between domains and even sub-domains. These expectations were generally met. Overall, sub-domains within attention, EF, perception, and memory, generally demonstrate recovery between 6–12 months, though basic processing speed recovers within a month, and working memory updating/tracking as early as 18 days, while verbal function demonstrated improvement within a month, but was likely unimpaired to start with.

Specifically, attention recovered by 12 months at the latest, with sub-domains recovering earlier: basic processing speed by a month (longer if more complex tasks used), divided attention by six months, and focused attention by 6–12 months. Attentional capacity was seemingly unimpaired, with some initial poor performance likely due to working memory updating/tracking aspects of tasks used. Executive functions recovered differentially, with working memory updating/tracking recovering by 18 days, response inhibition by six months, and decision-making demonstrating improvement by six months. Recovery of flexibility was inconsistent up to 12 months, while planning was still impaired at this stage. Verbal fluency was not indicated as impaired but did improve consistently nonetheless over the course of 12 months. Within learning and memory, STM recovered faster than LTM, as did verbal compared to visual. Verbal STM recovered by six months onwards, while verbal LTM took up to eight months. Visual STM recovered by eight months onwards, while visual LTM took 24 months. Perception and motor performance responded well to abstinence, with improvement or recovery generally indicated by six months, though with few studies synthesised, this is uncertain. Within this, perception of visual interference improved by 18 days, while perception (and fine motor skills) of simple figures recovered by six months, and perception of more complex designs took up to 12 months (similarly, emotional facial expression was still impaired at three months). Fine motor skills were likely unimpaired, but recovered by six months if so. In contrast to the displays of improvement or recovery, by 12 months, general concept formation and reasoning skills were consistently impaired, whilst recovery of sorting and shifting ability was inconsistent.

That most of the domains and sub-domains assessed were generally initially impaired (with the exception of verbal fluency and attentional capacity) supports previous research that has

found the same [5–7, 48], while differential recovery of executive functions supports that these are separable abilities [49]. When compared to Schulte, Cousijn [7], who also reviewed longitudinal literature, the current review similarly found recovery of verbal memory, response inhibition, and continued impairment of emotional facial expression recognition and planning. In contrast, Schulte, Cousijn [7] found no improvement in decision making, focused attention, or verbal functions, but did find improvement in reasoning ability. However, only two studies in Schulte, Cousijn [7] measured concept formation and reasoning (called Performance IQ in Schulte, Cousijn [7]), one was Manning, Wanigaratne [21], included in the current review, and the other Rosenbloom, Rohlfing [50], which was excluded here due to a very varied initial abstinence at baseline, ranging from three weeks to two years, as it was determined that it could not be considered an assessment of the impact of abstinence duration. Furthermore, only two studies in Schulte, Cousijn [7] assessed decision-making [22, 33], both of which are in the current review, while an additional study included here [19] was published later, giving further insight. Similarly, three studies in Schulte, Cousijn [7] assessed focused attention (called Sustained Attention), one among the seven included here [27], Rosenbloom, Rohlfing [50], excluded for the reasons described, and Sullivan, Rosenbloom [51], excluded from the current review due to using the DSM-III-R to identify AUD participants, which has good concordance with the DSM-IV and ICD-10 [52], but is slightly different [53] and so could capture a different group to more recent tools. Finally, verbal function (called Verbal IQ) was assessed by two studies in Schulte, Cousijn [7], Fujiwara, Brand [54], who examined ARBI patients (a different group), and Rosenbloom, Rohlfing [50], though it should be noted that the current review only described one study of verbal function, which is not at all conclusive.

In relation to neural correlates, inhibitory control, flexibility, working memory, planning, decision making, attention, reasoning, processing speed, verbal short- and long-term memory, are all arguably functions depending heavily on frontoparietal regions, particularly the PFC [55–59]. Exceptions include fine motor function which associates with frontal and cerebellar regions [60], perception which is frontoparietal and occipital [61], verbal fluency which is largely frontal [57], visual STM which is occipito-parietal, visual LTM which associates with the medial temporal lobe and hippocampus [62], vocabulary which involves frontal, temporal, thalamic, and cerebellar regions [63], and emotion recognition which associates with visual and limbic systems, prefrontal, temporoparietal, and subcortical areas such as the cerebellum [64].

The current findings of initial impairment therefore are in line with grey matter alterations in AUD which have been found consistently in parts of the PFC, anterior cingulate cortex, and insulae in AUD [2, 3, 65–68], in various frontal and parietal areas [2, 3, 67] and subcortical regions such as the thalamus, hippocampus and cerebellum [3, 67], along with widespread white matter reductions [1, 3] which are most pronounced in the frontal lobes, cerebellum, and limbic system [3]. Furthermore, that there were relatively consistent improvements, even to the point of recovery, of various sub-domains within attention, executive function, memory, and perception, supports findings of Zou, Durazzo [69] that the volumes of the anterior cingulate cortex, dorsolateral PFC, orbitofrontal cortex, and insula reached equivalent volume to controls by seven months of abstinence, while the hippocampal volume increased but still remained smaller than controls, perhaps explaining the slower recovery of visual LTM compared to other types of memory in this review. However, Durazzo, Mon [16] found that lobar and cerebellar brain volume increases associated with processing speed recovery only, not that of verbal or visual memory.

Relating to the impact of alcohol on the brain, impairment and recovery of many of these functions upon abstinence appears to support that the frontal lobe and cerebral cortex are

particularly vulnerable to damage by active AUD (the frontal lobe vulnerability, and whole brain hypotheses; Oscar-Berman and Marinkovic [70]), though it is likely that elements of cognitive function may be heritable and have a cyclical relationship with alcohol [71]. Given that the frontal lobes have rich connections with other brain regions, and that prefrontal functions are required for cognitive control, damage to this area may therefore influence performance on tests used for assessing functions of other brain regions [72]. Furthermore, given that Durazzo, Pennington [15] found that age independently predicted recovery on processing speed, attentional capacity, and memory, this supports the 'premature aging hypothesis', which posits that alcohol either ages the brain prematurely, or that age increases the vulnerability of the brain to alcohol [70, 73, 74].

With regards to the second aim of this review, various predictors were indicated for several domains/sub-domains. A consistent predictor of recovery was age, which predicted recovery of processing speed, attentional capacity, and memory [15], as did smoking status, and premorbid verbal intelligence (Vocabulary Test; Schmidt and Metzler [39]), while verbal memory was also predicted by education. Additionally, in smokers, visual short term memory recovery negatively associated with potentially pathological increases in brain volume [46]. A reduction of drinking behaviour in people important to the individual supported recovery of focused attention [29], while divided attention recovered more poorly with repeated cycles of withdrawal and relapse [33]. Finally, lower age of onset of first alcohol use and higher AUDIT score both associated with poorer recovery of a composite cognitive score [23], though more research is needed to understand how recovery of specific functions is impacted by these factors. Despite this, future research and intervention should try to consider all these predictive elements to ensure that best outcomes can be achieved for everyone.

## Robustness of synthesis

Study quality was assessed using the JBI Checklist for Cohort Studies (Moola, Munn [75], see Table 3). The majority (eleven) of studies were classed as 'good', allowing for relative trustworthiness of the current review, while three were considered 'moderate', and two were 'poor'. The most frequent issues throughout the literature reviewed were not including controls, confirming abstinence using self-report only, assessing function up to less than six months of abstinence (given that previous research suggested function may take up to a year to recover [6]), limited description of the characteristics of those lost to attrition, lack of strategies to reduce attrition bias, and not controlling for potential confounds in the statistical analysis. Furthermore, ultimately, as Schulte, Cousijn [7] stated, without studies that assess neuropsychological functioning before, during, and after AUD, there is less certainty about the findings with regards to the relationship of alcohol use to function at each of these stages. These issues should all be considered in future research.

Two of the 'moderate' quality studies [29, 46], and one of the 'poor' [25], did not compare to controls or normative data, and so were only considered as information on improvement within the review, not as direct assessment of recovery. Furthermore, as a safeguard to the quality of the review, despite Kaur, Sidana [25] reporting on predictors of cognitive recovery, these were not included, as the nature of statistical analyses and findings were unclear and contradictory throughout this source. Similarly, while Angerville, Ritz [23] (good) did include controls and normative data in their study design, this paper has been used here to infer improvement only (not recovery) due to a lack of comparison using tests of statistical significance at T0 to either normative data or to controls (despite indications of initial impairment). Improvement was measured via regression methods assessing the impact of abstinence duration, and therefore this has been synthesised.

**Table 3. Quality assessment using JBI checklist for cohort studies.**

| Author | 1. Were the two groups similar and recruited from the same population? | 2. Were the exposures measured similarly to assign people to both exposed and unexposed groups? | 3. Was the exposure measured in a valid and reliable way? | 4. Were confounding factors identified? | 5. Were strategies to deal with confounding factors stated? | 6. Were the groups/participants free of the outcome at the start of the study (or at the moment of exposure)? | 7. Were the outcomes measured in a valid and reliable way? | 8. Was the follow up time reported and sufficient to be long enough for outcomes to occur? | 9. Was follow up complete, and if not, were the reasons for loss to follow up described and explored? | 10. Were strategies to address incomplete follow up utilized? | 11. Was appropriate statistical analysis used? | Quality Score (%) Poor ≤ 49 Moderate = 50–69 Good ≥ 70 |
|---|---|---|---|---|---|---|---|---|---|---|---|---|
| Alhassoon et al., 2012 | Yes | Yes | Yes | Yes | Yes | | Yes | Yes | Yes | Yes | Yes | 100 |
| Angerville et al. 2023 | Yes | Yes | Yes | Yes | Yes | | Yes | Unclear | Yes | Yes | Yes | 90 |
| Bartels et al., 2007 | Not applicable | Not applicable | Yes | Yes | Yes | | Yes | Yes | Yes | Yes | No | 70 |
| Czapla et al., 2016 | Yes | Yes | Yes | Yes | Yes | | Yes | Yes | Yes | Yes | Yes | 100 |
| Cordovil De Sousa Uva et al., 2010 | Yes | Yes | Yes | Yes | Yes | | Yes | Unclear | No | No | Yes | 70 |
| Durazzo et al., 2014 | Not applicable | Not applicable | Yes | Yes | Yes | | Yes | Yes | Yes | Yes | Yes | 80 |
| Foisy et al., 2007 | Yes | Yes | Yes | Yes | Yes | | Yes | Unclear | Yes | No | Yes | 80 |
| Ioime et al., 2018 | Yes | Yes | Yes | Yes | Yes | | Yes | Yes | Yes | Yes | No | 90 |
| Kaur et al., 2020 | Not applicable | Not applicable | No | Yes | Yes | | Yes | Unclear | Unclear | Unclear | Unclear | 30 |
| Loeber et al., 2010 | Yes | Yes | Yes | Yes | Yes | | Yes | Yes | Yes | No | Yes | 90 |
| Manning et al., 2008 | Not applicable | Not applicable | Yes | Yes | Unclear | | Yes | Unclear | Unclear | Unclear | Yes | 40 |
| McCutcheon et al., 2016 | Not applicable | Not applicable | No | Yes | Yes | | Yes | Unclear | Yes | Yes | Yes | 60 |
| Petit et al., 2017 | Yes | Yes | Yes | Yes | Yes | | Yes | Unclear | Unclear | No | Yes | 70 |
| Pitel et al., 2009 | Yes | Yes | No | Yes | Yes | | Yes | Yes | Yes | Yes | No | 80 |
| Wegner et al., 2001 | Yes | Yes | Yes | Yes | Yes | | Yes | Unclear | Unclear | Unclear | Yes | 70 |

(*Continued*)

**Table 3.** (Continued)

| Author | 1. Were the two groups similar and recruited from the same population? | 2. Were the exposures measured similarly to assign people to both exposed and unexposed groups? | 3. Was the exposure measured in a valid and reliable way? | 4. Were confounding factors identified? | 5. Were strategies to deal with confounding factors stated? | 6. Were the groups/ participants free of the outcome at the start of the study (or at the moment of exposure)? | 7. Were the outcomes measured in a valid and reliable way? | 8. Was the follow up time reported and sufficient to be long enough for outcomes to occur? | 9. Was follow up complete, and if not, were the reasons for loss to follow up described and explored? | 10. Were strategies to address incomplete follow up utilized? | 11. Was appropriate statistical analysis used? | Quality Score (%) Poor $\leq$ 49 Moderate = 50–69 Good $\geq$ 70 |
|---|---|---|---|---|---|---|---|---|---|---|---|---|
| Yeh et al., 2007 | Not applicable | Not applicable | No | Yes | Yes | | Yes | Yes | No | Yes | Yes | 60 |

Note: due to the nature of this review, some checklist questions were not directly applicable. Therefore, question one was answered "Yes" if a control and AUD group were matched or described as similar on age, gender, or education. Question two was answered "Yes" if it was made clear that controls did not have a diagnosis of AUD. Question three was answered "Yes" if abstinence from alcohol was confirmed in AUD participants at each time, using additional methods to self-report (except for follow-ups conducted in inpatient settings where abstinence would be assured). Question six was not relevant as the outcome was neuropsychological change upon maintenance of abstinence in AUD rather than an incidence of illness, so this question was not used to assess study quality. Based on previous literature, question eight was answered "Yes" if the study followed up to six months of abstinence or longer. Question ten in the checklist suggested methods such as calculating person-years at risk, which again is not suitable for studies in this review, therefore this question was answered "Yes" if follow-up was complete, or if studies used statistical methods such as linear mixed modelling, multiple imputation, dummy variables, or sample weights. Alternatively, complete case analysis was deemed acceptable when characteristics of those lost to follow-up were like those who remained. Finally, question 11 was answered "Yes" if the statistical analysis used adjusted in some way for covariates/ confounds, or multiple dependent variables (if appropriate considering outcomes measured) including use of multivariate analysis or Bonferroni correction.

Manning, Wanigaratne [21] (poor) and Cordovil De Sousa Uva, Luminet [22] (moderate) however did include appropriate performance comparisons, and so any conclusions drawn because of their inclusion may need caution in interpretation. In particular, those relating to Manning, Wanigaratne [21], such as within planning, verbal fluency, and verbal function, for which this was the only study at all to assess comparative to normative data. This study was also at odds with other papers regarding absence of initial impairment for processing speed, working memory updating/tracking, visual concept formation, verbal STM, and fine motor function; however as multiple studies opposed this, these findings were considered the exception. Similarly, most sub-domains and domains were assessed by three or more studies, except for planning, verbal fluency, and verbal function (which are also those contributed to most significantly by Manning, Wanigaratne [21]), suggesting that all functions except these, the synthesis results are trustworthy regarding the quantity and quality of evidence.

A strength of this is review is the grouping of tasks under multiple functions (as opposed to Crowe, Cammisuli [5], Stavro, Pelletier [6]), as it is recognised that multiple tasks span various domains and sub-domains [7], and indeed that specific elements of each task may measure different abilities (as described in Table 2). Additionally, the review considers the predictive features contributing to the variance in recovery, though a limitation is the inability to control for confounds such as practice effects, medication, and comorbid health problems, particularly comorbid psychiatric or gastrointestinal and liver disorders (such as cirrhosis) which are highly comorbid in AUD [76, 77] and along with the related treatment (e.g., psychotropic medications) also impact cognitive function [78–80]. the reality is that in populations recovering from AUD, medication and comorbid health problems are likely to be the norm [77], suggesting that these are less isolated contributions to variance, though liver disease and psychotropic medications would be useful criteria to exclude where feasible.

The authors used various methods to strengthen the quality of this narrative synthesis by reducing bias, including pre-registering the protocol with PROSPERO, publishing the protocol so as to gain valuable feedback prior to conducting the review [11], limiting to only DSM-IV/5 and ICD-10/11 to try to ensure comparative cohorts [81], full-text screening being checked by multiple reviewers, and 10% of quality assessment being independently conducted.

## Future research and implications for practice

Future research into neuropsychological recovery from AUD should consider the impacts of age, premorbid intelligence, smoking, education, brain volume, number of previous treatments, and drinking behaviour of those close to the individual, as all were identified as predictors [15, 29, 33], so could explain some of the variance identified across the literature. Large scale prospective studies are also needed to understand what functional differences compared to controls may be pre-existing, and how this may be cyclical, as some are likely to have causal relationships with AUD [82, 83].

Interestingly, in the RVP task, which required a combination of multiple executive or executive and reasoning abilities, initial and continued impairment was more likely than in tasks that were more function specific. This indicates that to capture impairment, the additive nature of task requirements should be considered in future research, particularly as day-to-day demands are likely to be competing for cognitive resource in this way. Indeed, in the development of their virtual reality 'Jansari assessment of Executive Functions', Jansari, Devlin [84] suggested that assessing individual EF is like a conductor listening to individual instruments of an orchestra to decide if they work together, when listening to them in harmony would give a more ecologically sound understanding of ability. Additionally, in a study of a dual working memory updating (N-back) and response inhibition (Flanker) task, Kim, Wittenberg [55]

found that increasing the working memory load led to a decrease in both working memory and response inhibition performance, indicating an interaction effect of the two EFs on individual task performance. Furthermore, Finn, Justus [85] suggest that individuals with low working memory capacity are more susceptible to a reduction in inhibitory control due to alcohol, suggesting a complex interaction between alcohol use and the additive impact of cognitive functions, which should be further investigated in the context of recovery. In addition, there was some evidence that certain verbal functions recover faster than visual (specifically relating to processing speed, flexibility, and memory), which is something that should be studied further, as it may account for some of the discrepancies within the literature. Of the studies reviewed here, there were large discrepancies between the timeline of assessment, which weakens the comparisons that can be made. To combat this, future studies should seek to capture regular follow-up assessments up to and beyond at least a year of abstinence.

Importantly, this review indicates that several functions do recover with prolonged abstinence from alcohol in individuals with AUD. This information may be encouraging for patients, and therefore could be provided to them by practitioners in a plain language format (e.g., in the form of motivational interviewing techniques), to help them engage in and maintain recovery behaviours. The impact of this could also be investigated, as to whether knowledge of this information improves maintenance of abstinence.

## Conclusions

The findings of this systematic review suggest that sub-domains within attention, executive function, perception, and memory, demonstrate recovery, generally between 6–12 months. This supports both the frontal lobe vulnerability and whole brain hypotheses of alcohol damage, while the consistent finding of age as a predictor of function recovery supports the premature aging hypothesis. Overall robustness of results was deemed good, though not for planning, verbal fluency, and verbal function, for which further research addressing previous methodological limitations is required, which include lack of control groups, additional methods to self-report to confirm abstinence, description/control for attrition, statistical control of confounds, and of long enough study durations to capture change. Future research should consider the impacts of identified predictors, as these may explain some of the variance across the literature. Large scale prospective studies will develop our understanding of what functional differences may be pre-existing or cyclical. Finally, the additive nature of task requirements in cognitive recovery should be investigated, particularly as day-to-day demands are likely to be layered. With regards to implications for practice, practitioners should consider delivering to patients the finding that cognitive recovery in several functions can be achieved with abstinence, as this may provide encouragement and support positive behaviour change. The impact of such delivery on maintenance of abstinence could also be assessed.

## Supporting information

**S1 Table. PRISMA 2020 abstracts checklist.**
(PDF)

**S2 Table. PRISMA 2020 checklist.**
(PDF)

**S3 Table. Systematic search strategies for APA PsycInfo, EBSCO MEDLINE, CINAHL, and Web of Science.**
(PDF)

S4 Table. Papers excluded at the full-text screening stage, with reason for exclusion. (PDF)

## Author Contributions

**Conceptualization:** Anna Powell, Harry Sumnall, Catharine Montgomery.

**Formal analysis:** Anna Powell.

**Investigation:** Anna Powell.

**Methodology:** Anna Powell, Harry Sumnall, Catharine Montgomery.

**Project administration:** Anna Powell.

**Supervision:** Harry Sumnall, Catharine Montgomery.

**Validation:** Jessica Smith, Rebecca Kuiper.

**Writing – original draft:** Anna Powell.

**Writing – review & editing:** Harry Sumnall, Jessica Smith, Rebecca Kuiper, Catharine Montgomery.

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
