## [Decision Letter · Decision Letter 0]

20 Sep 2023

PONE-D-23-18208Recovery of neuropsychological function following abstinence from alcohol in adults diagnosed with an Alcohol Use Disorder: Systematic review of longitudinal studiesPLOS ONE

Dear Dr. Powell,

Thank you for submitting your manuscript to PLOS ONE. After careful consideration, we feel that it has merit but does not fully meet PLOS ONE’s publication criteria as it currently stands. Therefore, we invite you to submit a revised version of the manuscript that addresses the points raised during the review process.

We look forward to receiving your revised manuscript.

Kind regards,

Valerio Manippa

Academic Editor

PLOS ONE

Reviewers' comments:

Reviewer's Responses to Questions

**Comments to the Author**

1. Is the manuscript technically sound, and do the data support the conclusions?

Reviewer #1: Yes

Reviewer #2: Yes

Reviewer #3: Yes

2. Has the statistical analysis been performed appropriately and rigorously? 

Reviewer #1: N/A

Reviewer #2: N/A

Reviewer #3: Yes

3. Have the authors made all data underlying the findings in their manuscript fully available?

Reviewer #1: Yes

Reviewer #2: Yes

Reviewer #3: Yes

4. Is the manuscript presented in an intelligible fashion and written in standard English?

Reviewer #1: Yes

Reviewer #2: Yes

Reviewer #3: Yes

5. Review Comments to the Author

Reviewer #1: Thank you for this important and well conducted review. The most important aspect is that cognitive functions can be restored in people who are willing and ready to achieve abstinence from alcohol. Some recommendations for clinicians to include these information in motivational interviews for patients making sure they avoid any judgement and stigmatising language and a plain language version for alcohol service users to be aware that some of their difficulties can be overcome and full functionality recuperated would increase the already good contribution of this paper.

I wonder whether an additional paragraph with implications for research and for practices could be added to this paper.

Reviewer #2: This review aim to synthetis changes in neuropsychological performances in patients with AUD. It notably shows that approximately the majority of the patients who were initially impaired at baseline no longer exhibited any deficit at follow-up. By doing so, the authors address a central matter in the field of AUD, as identifying the recovery of cognitive deficits

While I recognize the importance of such explorations, and acknowledge some of the strengths of this review, and notably its willingness to control more confounding factors compared to previous similar work, I also have a few comments and remarks that I believe may help to improve the paper before it can be considered for publication. These remarks especially concern methodological aspects that may render some results potentially misleading in their current form.

Major comments

First of all, the authors should highlight the originality of this review and the added value, compared to previous reviews in the introduction section (notably, stravo et al, 2012 Addic Biol).

Secondly, the exact time of recovery had not been systematically reported in the results section of the manuscript or at least in the discussion. I know that the time points of the follow up assesment are indicated in tables and figures, but it would be useful to readers to be aware of the Many discrepancies between the studies regarding the exact time of recovery, (ex: range of recovery ?). It would be interesting at least to discuss this aspect, and to provide hypothesis to improve future studies in this field.

Finally, the authors have correctly noticed that many cofounders have been included in the analysis of cognitive impairement. However important limitations can be added to the exclusion criteria of the studies notably the use of psychotropic medications or the presence of liver disease such as cirrhosis , that may impaired cognitive assesment of patients with AUD. Please add this subject in the discussion section.

Minor comments

Can you explain why studies such as Mulhauser et al., 2018, Luqiens 2019, Angerville et al. 2023...were not included in the review ?

page 4, line 77 " A comparator was required, in the form of adults aged 18-24 without AUD"

Why the comparator (control group was aged 18-24 while the inclusion criteria was AUD patient aged 18-54 ?

can you explain why the comparator (control group) should be used at least at baseline ? it would be interesting regarding the aim of the review to include studies using compartors at follow up comparaison also?

why a metaanalysis could not be performed ?

Reviewer #3: This work addresses the question of recovery of neuropsychological function following abstinence from AUD through a systematic review of longitudinal studies.

The systematic review was conducted from 1999 with a review protocol previously published (Powel et al., 2022).

The analysis is detailed and carefully carried out, but there is a lack of synthesis, which reduces the effectiveness of the message to the reader. This paper should be improved by clarifying and concisely presenting the main findings.

Indeed, the authors refer to a previous study (Powell et al. 2022) to justify the editorial brevity (Introduction, Objectives, Methods). This does not necessarily explain the lack of drafting (see Introduction) and deserves clarification.

Especially, the brief introduction fails to present the context of the research, whether it is the scientific question about recovery of neuropsychological function after abstinence or the methodology used for the systematic review in the context of previous research in this area. Whilst the authors' would like us to refer back to their previous publication, this is not clearly stated in the introduction and requires further clarification of the context. The interest of the subject is not justified.

The objectives are strictly identical to the earlier review by Powell et al. (2022) (6% of the document has been flagged for potential plagiarism by the anti-plagiarism software), however, it is necessary for the authors to provide a better explanation of how they ensured continuity with the previous publication. The aim of this review should explicitly mention that the outcomes presented are a result of the review and not any experimental or observational research. Objectives should be presented in more concrete terms.

The results are then presented by neuropsychological function. To make this section clearer, a presentation of main outcomes would be welcome.

The headings in this section should be adapted.

For the sake of completeness, the authors enumerate findings from the literature in the Discussion section to the detriment of the meaning and message to be conveyed to the reader. In order to draw more precise conclusions, it would be beneficial to complete comparisons with other studies and discuss your findings in relation to the existing literature (for example, see line 466 to 472). Likewise, the overall message to the reader would be clarified by a synthesis of the results and a presentation of the limitations of the literature.

Figures 1 and 2 have no title or legend.

References are not all in the same format. They should be harmonized.

6. PLOS authors have the option to publish the peer review history of their article (what does this mean?). If published, this will include your full peer review and any attached files.

Reviewer #1: **Yes: **Marica Ferri

Reviewer #2: **Yes: **Angerville Bernard

Reviewer #3: **Yes: **Judith André

---

## [Author Response · Author response to Decision Letter 0]

4 Nov 2023

Dear Professor Emily Chenette,

Thank you for giving us the opportunity to submit a revised draft of our manuscript titled ‘Recovery of neuropsychological function following abstinence from alcohol in adults diagnosed with an Alcohol Use Disorder: Systematic review of longitudinal studies’ to PLOS ONE (Manuscript ID: PONE-D-23-18208). We appreciate the time and effort that you and the reviewers have dedicated to providing valuable feedback on our manuscript. We are grateful for the insightful comments on our paper. We have been able to incorporate changes that reflect most of the suggestions provided. 

We look forward to hearing from you in due time regarding our submission and to respond to any further questions and comments you may have.

Sincerely,

Anna Powell

Here is a point-by-point response to the reviewers’ comments and concerns.

Reviewer 1:

1. Thank you for this important and well conducted review.

Response: We thank Reviewer 1 for the positive comment. 

2. The most important aspect is that cognitive functions can be restored in people who are willing and ready to achieve abstinence from alcohol. Some recommendations for clinicians to include these information in motivational interviews for patients making sure they avoid any judgement and stigmatising language and a plain language version for alcohol service users to be aware that some of their difficulties can be overcome and full functionality recuperated would increase the already good contribution of this paper. I wonder whether an additional paragraph with implications for research and for practices could be added to this paper.

Response: Reviewer 1 raises a good point. We had originally given some information on future research in the conclusion, however, agree that an earlier section with more detail on both this and implications for practice, would be better. We have therefore added this (with the brilliant suggestion by Reviewer 1 of motivational interviewing techniques to provide this information to patients), and have adapted the conclusion accordingly. Location(s): lines 702-736, 746-759

Reviewer 2: 

1. This review aims to synthesise changes in neuropsychological performances in patients with AUD. It notably shows that approximately the majority of the patients who were initially impaired at baseline no longer exhibited any deficit at follow-up. By doing so, the authors address a central matter in the field of AUD, as identifying the recovery of cognitive deficits. While I recognize the importance of such explorations, and acknowledge some of the strengths of this review, and notably its willingness to control more confounding factors compared to previous similar work, I also have a few comments and remarks that I believe may help to improve the paper before it can be considered for publication. These remarks especially concern methodological aspects that may render some results potentially misleading in their current form.

Response: We thank Reviewer 2 for their positive feedback, and for their useful critique. 

1. First of all, the authors should highlight the originality of this review and the added value, compared to previous reviews in the introduction section (notably, stravo et al, 2012 Addic Biol).

Response: We agree, and so in response to comments by Reviewer 2 and 3, we have rewritten the introduction to better justify the review, particularly by highlighting the importance of reviewing longitudinal work and grouping tasks under multiple function domains and sub-domains. We have also directed readers to the published protocol for more information on the justification of the review - due to the length of the paper, we are hesitant to extend it further by lengthening the introduction more, though if Reviewer 2 feels strongly about this we are happy to further revise the introduction. For clarity, we have also highlighted in the methods section that we grouped tasks across multiple domains. Location(s): lines 52-75, 74-75, 117-119

2. Secondly, the exact time of recovery had not been systematically reported in the results section of the manuscript or at least in the discussion. I know that the time points of the follow up assesment are indicated in tables and figures, but it would be useful to readers to be aware of the Many discrepancies between the studies regarding the exact time of recovery, (ex: range of recovery ?). It would be interesting at least to discuss this aspect, and to provide hypothesis to improve future studies in this field.

Response: We appreciate this insight by Reviewer 2. We have updated the manuscript to include range of follow-up time for each function in the relevant narrative synthesis sections of the result section. We have also suggested that future research seeks to capture regular follow-up assessments up to and beyond at least a year of abstinence. Location(s): lines 729-730

3. Finally, the authors have correctly noticed that many cofounders have been included in the analysis of cognitive impairement. However important limitations can be added to the exclusion criteria of the studies notably the use of psychotropic medications or the presence of liver disease such as cirrhosis , that may impaired cognitive assesment of patients with AUD. Please add this subject in the discussion section.

Response: Thank you for this comment. We had acknowledged in the Robustness of Synthesis section that the inability to control for confounds such as practice effects, medication, and comorbid health problems, was a limitation of the review. However, in accordance with your comment, we have now highlighted within this that specifically cirrhosis and psychiatric disorders are highly comorbid and impact cognitive function, and that the related treatment (e.g., psychotropic medications) can also impact function, and that these should be excluded where feasible. Location(s): lines 690-696

4. Can you explain why studies such as Mulhauser et al., 2018, Luqiens 2019, Angerville et al. 2023...were not included in the review ?

Response: We appreciate Reviewer 2 for bringing these studies to our attention, as two of them were not returned by our search strategy. One of these two has now been included in the review. Of the three studies:

Mulhauser et al. (2018) and Luquiens et al. (2019) do not meet inclusion criteria for the review, due to their including several patients with a comorbid substance use disorder (cannabis use disorder, and cocaine use disorder) as part of their AUD sample. Our review criteria states that for papers in which participants are reported as consuming other substances as well as alcohol, alcohol must be the primary substance, which is not the case where the disorders are described as comorbid. This could confound the results of the papers, and subsequently the review synthesis, no other papers included report on AUD participants with comorbid substance use disorders. An exception in the criteria (described in the protocol) is the SUD tobacco use disorder, due to its particularly high comorbidity with AUD. This has been made clearer in the review Method section (lines 98-101). Additionally, Luquiens et al., (2019) only recruited patients with a MoCA score of <26, which is a screening indicator for ARBI, and would have excluded this study alone.

Angerville et al. (2023) had not previously been returned as part of our search strategy, and as this paper excluded participants with a history or current use of other substances, and meets the rest of the inclusion criteria, it has been added to the review. We thank Reviewer 2 for this addition. Furthermore, in the process of adding this paper, we realised there had previously been an issue with some incorrect value recording in the PRISMA flow-chart, which has now been fixed.

5. page 4, line 77 " A comparator was required, in the form of adults aged 18-24 without AUD". Why the comparator (control group was aged 18-24 while the inclusion criteria was AUD patient aged 18-54 ?

We thank Reviewer 2 for bringing this to our attention, this was a typo, and has been corrected, both sentences now state “adults aged 18-64”. Location(s): line 94

6. can you explain why the comparator (control group) should be used at least at baseline ? it would be interesting regarding the aim of the review to include studies using compartors at follow up comparaison also?

Response: Seven of the studies included in this review did include control group comparison at one or more follow-up assessments, Table 1 column ‘Comparator’ gives more clarification on this, and it is also described in the first paragraph of the results. Of the eleven studies which compared cognitive function in recovery to controls, only four did so once. Location(s): Table 1, lines 125-128

7. why a metaanalysis could not be performed?

Response: Due to the breadth of cognitive functions assessed, there was significant heterogeneity across the studies included in this review, which meant that meta-analysis was not suitable. This is touched on briefly in the Method section (line 114), and was anticipated when we designed the review protocol as a planned narrative synthesis (see Powell et al. (2022)).

As we knew we would not be able to conduct meta-analysis, we ensured that our narrative synthesis strategy was well-planned in accordance with Popay et al. (2006) and the University of York’s Centre for Reviews and Dissemination (2009), using multiple grouping of tasks (Schulte et al., 2014) and the well-known Lezak et al. (2012) book Neuropsychological Assessment as guidance for the grouping process, and with the matrix to provide a visual basis for the written synthesis. As our protocol was published, we also ensured that the narrative synthesis methods received peer review and approval prior to beginning this process. 

Reviewer 3: 

1. This work addresses the question of recovery of neuropsychological function following abstinence from AUD through a systematic review of longitudinal studies. The systematic review was conducted from 1999 with a review protocol previously published (Powel et al., 2022). The analysis is detailed and carefully carried out, but there is a lack of synthesis, which reduces the effectiveness of the message to the reader. This paper should be improved by clarifying and concisely presenting the main findings.

Response: We thank Reviewer 3 for their time providing thoughtful feedback on our paper, and for their positive comment on our attention to detail. In response to another of Reviewer 3’s comments, we have clarified the main findings in the first paragraph of the results, and added a more detailed summary of the synthesis to the discussion, to improve the overall synthesis of the review. Location(s): lines 131-134, 547-570

2. Indeed, the authors refer to a previous study (Powell et al. 2022) to justify the editorial brevity (Introduction, Objectives, Methods). This does not necessarily explain the lack of drafting (see Introduction) and deserves clarification.

Response: We agree, and so in response to comments by Reviewer 2 and 3, we have rewritten the introduction to better justify the review. We have also directed readers to the published protocol for more information on the justification of the review - due to the length of the paper, we are hesitant to extend it further by lengthening the introduction more, though if Reviewer 3 feels strongly about this we are happy to further revise the introduction. Location: lines 55-75, 

Regarding review methods, there is a sentence directing readers to the published protocol in the Methods section if they wish to read a more detailed description. Location: lines 81-83

3. Especially, the brief introduction fails to present the context of the research, whether it is the scientific question about recovery of neuropsychological function after abstinence or the methodology used for the systematic review in the context of previous research in this area. Whilst the authors' would like us to refer back to their previous publication, this is not clearly stated in the introduction and requires further clarification of the context. The interest of the subject is not justified. 

Response: Please see response to Reviewer 3 comment 2. 

4. The objectives are strictly identical to the earlier review by Powell et al. (2022) (6% of the document has been flagged for potential plagiarism by the anti-plagiarism software), however, it is necessary for the authors to provide a better explanation of how they ensured continuity with the previous publication.

Response: The objectives for this review are consistent with those of the protocol because it was adhered to consistently. We agree that this may not have been clear, and have made a note in the methods section to this effect. Location: line 81

5. The aim of this review should explicitly mention that the outcomes presented are a result of the review and not any experimental or observational research. Objectives should be presented in more concrete terms.

Response: We have added a statement to this effect to the Objective(s) section, however as our objectives follow those of the published protocol, we do not feel it would be appropriate to deviate from this. Location: line 77

6. The results are then presented by neuropsychological function. To make this section clearer, a presentation of main outcomes would be welcome. The headings in this section should be adapted.

Response: We agree that there was a lack of clarity regarding main findings in this section, and have added to the first paragraph of the results highlighting these (line 131-134). Headings within the results relate to the function groupings, which were informed by the well-known Lezak et al. (2012) book Neuropsychological Assessment, which we feel is the best way to present this information. It is also how we proposed we would group, describe, and discuss data in the synthesis, in our published proposal, so do not wish to deviate from this if we can avoid it.

7. For the sake of completeness, the authors enumerate findings from the literature in the Discussion section to the detriment of the meaning and message to be conveyed to the reader. In order to draw more precise conclusions, it would be beneficial to complete comparisons with other studies and discuss your findings in relation to the existing literature (for example, see line 466 to 472).

Response: We thank Reviewer 3 for this comment, we agree and have added a more in-depth exploration of the results, specifically in comparison to Schulte et al. (2014) whose review methodology was most like ours. We have also highlighted that the differential recovery of executive functions supports literature suggesting that these are separable abilities. We hope the discussion is now acceptable to Reviewer 3. Location: lines 573-593

8. Likewise, the overall message to the reader would be clarified by a synthesis of the results and a presentation of the limitations of the literature.

Response: As mentioned, we have made several changes to improve the synthesis of the review, including clarifying the main findings in the results section (lines 131-134), adding a more detailed summary of the synthesis to the discussion (lines 547-570), and adding a more in-depth view of how this work sits in relation to previous literature (lines 573-593). In response to a comment by Reviewer 1, we have also added a section on Future Research and Implications for Practice (lines 702-736), which we hope will also improve the overall message to the reader. Limitations of the literature are addressed in the Robustness of Synthesis section (lines 640-675).

9. Figures 1 and 2 have no title or legend.

Response: Thank you for this comment. We have adhered to PLOS ONE guidelines, in which figure titles are inserted immediately in-text after the first paragraph in which the figure is cited, but figure files are uploaded separately. Therefore, the figure files do not contain the titles, but the manuscript itself does. 

10. References are not all in the same format. They should be harmonized.

Our thanks to Reviewer 3 for bringing this to our attention, upon checking, we realised that there had been an issue with the DOI format being different between references, which we have rectified. 

Angerville, B., Ritz, L., Pitel, A.-L., Beaunieux, H., Houchi, H., Martinetti, M. P., . . . Dervaux, A. (2023). Early Improvement of Neuropsychological Impairments During Detoxification in Patients with Alcohol Use Disorder. Alcohol and Alcoholism, 58(1), 46-53. https://doi.org/10.1093/alcalc/agac048

Centre for Reviews and Dissemination. (2009). Systematic reviews: CRD's guidance for undertaking reviews in healthcare. York: University of York NHS Centre for Reviews & Dissemination. 

Lezak, M. D., Howieson, D. B., Bigler, E. D., & Tranel, D. (2012). Neuropsychological assessment, 5th ed. Oxford University Press. 

Luquiens, A., Rolland, B., Pelletier, S., Alarcon, R., Donnadieu-Rigole, H., Benyamina, A., . . . Perney, P. (2019). Role of patient sex in early recovery from alcohol-related cognitive impairment: women penalized. Journal of Clinical Medicine, 8(6), 790. https://doi.org/10.3390/jcm8060790

Mulhauser, K., Weinstock, J., Ruppert, P., & Benware, J. (2018). Changes in neuropsychological status during the initial phase of abstinence in alcohol use disorder: neurocognitive impairment and implications for clinical care. Substance Use & Misuse, 53(6), 881-890. https://doi.org/10.1080/10826084.2017.1408328

Popay, J., Roberts, H., Sowden, A., Petticrew, M., Arai, L., Rodgers, M., . . . Duffy, S. (2006). Guidance on the conduct of narrative synthesis in systematic reviews. A product from the ESRC methods programme Version, 1, b92. https://doi.org/10.13140/2.1.1018.4643

Powell, A., Sumnall, H., Smith, J., Kuiper, R., & Montgomery, C. (2022). Recovery of neuropsychological function following abstinence from alcohol in adults diagnosed with an alcohol use disorder: Protocol for a systematic review of longitudinal studies. PLOS ONE, 17(9), e0274752. https://doi.org/10.1371/journal.pone.0274752

Schulte, M. H. J., Cousijn, J., den Uyl, T. E., Goudriaan, A. E., van den Brink, W., Veltman, D. J., . . . Wiers, R. W. (2014). Recovery of neurocognitive functions following sustained abstinence after substance dependence and implications for treatment. Clinical Psychology Review, 34(7), 531-550. https://doi.org/10.1016/j.cpr.2014.08.002

---

## [Decision Letter · Decision Letter 1]

5 Dec 2023

Recovery of neuropsychological function following abstinence from alcohol in adults diagnosed with an Alcohol Use Disorder: Systematic review of longitudinal studies

PONE-D-23-18208R1

Dear Dr. Powell,

We’re pleased to inform you that your manuscript has been judged scientifically suitable for publication and will be formally accepted for publication once it meets all outstanding technical requirements.

Kind regards,

Valerio Manippa

Academic Editor

PLOS ONE

Additional Editor Comments (optional):

Reviewers' comments:

Reviewer's Responses to Questions

**Comments to the Author**

1. If the authors have adequately addressed your comments raised in a previous round of review and you feel that this manuscript is now acceptable for publication, you may indicate that here to bypass the “Comments to the Author” section, enter your conflict of interest statement in the “Confidential to Editor” section, and submit your "Accept" recommendation.

Reviewer #2: All comments have been addressed

2. Is the manuscript technically sound, and do the data support the conclusions?

Reviewer #2: Yes

3. Has the statistical analysis been performed appropriately and rigorously? 

Reviewer #2: Yes

4. Have the authors made all data underlying the findings in their manuscript fully available?

Reviewer #2: Yes

5. Is the manuscript presented in an intelligible fashion and written in standard English?

Reviewer #2: Yes

6. Review Comments to the Author

Reviewer #2: (No Response)

7. PLOS authors have the option to publish the peer review history of their article (what does this mean?). If published, this will include your full peer review and any attached files.

Reviewer #2: **Yes: **Bernard Angerville

---

## [Editor Report · Acceptance letter]

21 Dec 2023

PONE-D-23-18208R1 

PLOS ONE

Dear Dr. Powell, 

I'm pleased to inform you that your manuscript has been deemed suitable for publication in PLOS ONE. Congratulations! Your manuscript is now being handed over to our production team.

Kind regards, 

on behalf of

Dr. Valerio Manippa 

Academic Editor

PLOS ONE